# T2I-R1:
# Reinforcing Image Generation with Collaborative Semantic-level and Token-level CoT

**Dongzhi Jiang**[*1], **Ziyu Guo**[*2], **Renrui Zhang**[*†1✉], **Zhuofan Zong**[1], **Hao Li**[3]
**Le Zhuo**[1,3], **Shilin Yan, Pheng-Ann Heng**[2], **Hongsheng Li**[1,3,4✉]

[1]CUHK MMLab    [2]CUHK IMIXR    [3]Shanghai AI Laboratory    [4]CPII under InnoHK

{dzjiang, ziyuguo, renruizhang}@link.cuhk.edu.hk
hsli@ee.cuhk.edu.hk

[*]Equal Contribution    [†]Project Leader    [✉]Corresponding author

## Abstract

Recent advancements in large language models have demonstrated how chain-of-thought (CoT) and reinforcement learning (RL) can improve performance. However, applying such reasoning strategies to the visual generation domain remains largely unexplored. In this paper, we present **T2I-R1**, a novel reasoning-enhanced text-to-image generation model, powered by RL with a bi-level CoT reasoning process. Specifically, we identify two levels of CoT that can be utilized to enhance different stages of generation: (1) the semantic-level CoT for high-level planning of the prompt and (2) the token-level CoT for low-level pixel processing during patch-by-patch generation. To better coordinate these two levels of CoT, we introduce **BiCoT-GRPO** with an ensemble of generation rewards, which seamlessly optimizes both generated CoTs within the same training step. By applying our reasoning strategies to the baseline model, Janus-Pro, we achieve superior performance with 13% improvement on T2I-CompBench and 19% improvement on the WISE benchmark, even surpassing the state-of-the-art model FLUX.1. All the training code and data are available at https://github.com/CaraJ7/T2I-R1.

## 1 Introduction

The emergence of advanced Large Language Models (LLMs) [62, 64, 81, 95], such as OpenAI o1 [65] and DeepSeek-R1 [20], has demonstrated considerable reasoning capabilities across domains including mathematics [1, 27, 55, 54] and coding [8, 30, 53]. Through reinforcement learning (RL) [72, 73, 52], these models analyze problems progressively with a comprehensive Chain-of-Thought (CoT) [84, 36, 24, 32, 105, 23] before providing answers, significantly enhancing output accuracy.

The CoT reasoning strategies have also been extended to the visual domain. Recent Large Multi-modal Models (LMMs) [7, 59, 103, 92] have adapted the paradigm to accommodate the visual understanding task [51, 105, 32, 22]. These advanced LMMs can jointly process images and their associated textual queries, performing step-by-step analyses of visual details and integrating them with reasoning steps to derive final answers. Concurrently, CoT-like reasoning has been initially investigated in the visual generation task [70, 99, 74, 37, 35, 10, 34, 100], particularly in autoregressive text-to-image generation. The pioneering work, *'Image Generation with CoT'* [24], regards the progressive generation of the image tokens as a kind of CoT analogous to that of the text tokens, and proposes to optimize this intermediate process to enhance the image quality.

39th Conference on Neural Information Processing Systems (NeurIPS 2025).

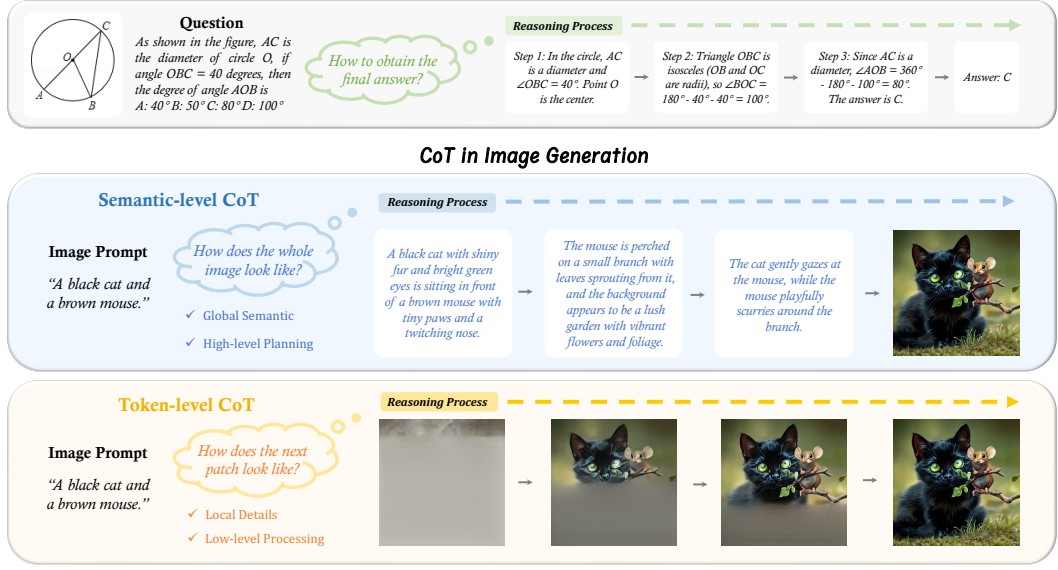

Figure 1: **The Illustration of CoT in Image Understand and Generation Tasks.** In the image understanding task, the CoT is the textual reasoning process. In the autoregressive visual generation task, we identify two levels of CoT: the semantic-level and token-level CoT. The semantic-level CoT is the high-level planning prior to the image generation, in the form of text. The token-level CoT is the intermediate patch-by-patch generation process, focusing on the local pixel details within a patch, in the form of image tokens.

Despite these advances, the exploration of CoT for image generation remains preliminary. Unlike image understanding, image generation requires the complex interpretation of cross-modal alignment and the synthesis of fine-grained visual details. To address these challenges, we identify two distinct levels of CoT reasoning that can be leveraged to enhance image generation, as illustrated in Fig. 1:

- **Semantic-level CoT** is the textual reasoning about the image to generate, which is introduced prior to the image generation. The semantic-level CoT designs the global structure of the image, e.g., the appearance and location of objects. In case the prompt requires reasoning shown in Fig. 2, the semantic-level CoT also helps to deduce the objects to generate. Optimizing the semantic-level CoT could explicitly decouple the planning and reasoning of the prompt from the subsequent image tokens generation, making the generation easier.

- **Token-level CoT** is the intermediate patch-by-patch generation process of the image, as originally introduced in [24]. This process could be viewed as a form of CoT as it outputs each subsequent token conditioned on all previous tokens within a discrete space, similar to the textual CoT. Unlike semantic-level CoT, token-level CoT focuses on low-level details like pixel generation and maintaining visual coherence between adjacent patches. Optimizing the token-level CoT can enhance both the generation quality and the alignment between the prompt and the resulting images.

Despite recognizing these two levels of CoT, a critical question remains unaddressed: *How can we enhance and coordinate them for text-to-image generation?* Current mainstream generative models [76, 79, 70, 37] are trained exclusively on generation targets, lacking the explicit textual understanding required for semantic-level CoT reasoning. Although introducing a separate model (e.g., an LLM) specifically for prompt interpretation [13] is technically feasible, this approach would significantly increase computational costs, complexity, and deployment challenges. Recently, a trend has arisen to merge visual understanding and generation within a single model. Building upon LMMs, these unified LMMs (ULMs) [86, 93, 107, 9] could not only understand the visual inputs but also generate images from text prompts. However, their two capabilities are still decoupled, typically pre-trained in two independent stages, with no clear evidence that the understanding capabilities can

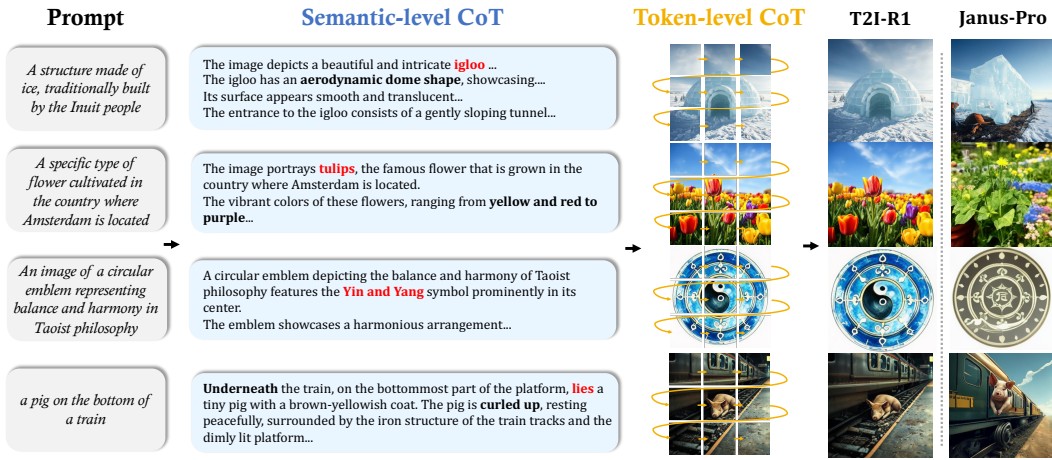

| Prompt | Semantic-level CoT | Token-level CoT | T2I-R1 | Janus-Pro |
|---|---|---|---|---|

Figure 2: **Visualization of the Image Generation Process of T2I-R1.** All the prompts need reasoning or contain an uncommon scenario. We observe that T2I-R1 successfully deduces the true intention behind the prompt or provides a sensible imagination (highlighted in the text) to produce a satisfying result compared with the baseline model, Janus-Pro.

benefit generation. Given these potentials and issues, we start from a ULM and enhance it to unite both the semantic-level and token-level CoT into one framework for text-to-image generation.

To fulfill our target, we introduce **BiCoT-GRPO**, an RL method to jointly optimize the two levels of CoT for ULM. We opt for RL instead of supervised fine-tuning (SFT) for two reasons: First, the ULM has possessed the fundamental ability needed for the semantic-level and token-level CoT; our goal is only to elicit the fusion of these two abilities by guiding the model's self-exploration. Second, RL methods have proven highly effective for enhancing reasoning capabilities, which are essential for both levels of CoT. Specifically, we first instruct the ULM to imagine and plan the image based on the prompt to obtain the semantic-level CoT. Then, we feed it into the ULM as the condition for the subsequent image generation for token-level CoT. We simultaneously generate multiple images from each prompt and then compute group-relative reward to optimize both levels of CoT within the same iteration. Unlike understanding tasks, where clearly defined rules for rewards exist, image generation lacks such standardized rules. Therefore, we propose to utilize an ensemble of diverse vision experts [90, 82, 49, 24] as reward models. This reward design serves two critical purposes: it evaluates generated images from multiple dimensions to ensure reliable quality assessment, while also functioning as a regularization method to prevent the ULM from hacking a single reward model.

Through the proposed reasoning strategies, we obtain **T2I-R1**, the first reasoning-enhanced text-to-image model combining the semantic-level and token-level CoT. Empirical results show that our approach outperforms baseline models by 13% and 19% improvements on the T2I-CompBench and WISE benchmark, and even surpasses the previous state-of-the-art model FLUX.1. Qualitative analysis reveals that our method empowers the model to generate more human-aligned results by reasoning about the true intentions behind the prompt and demonstrates enhanced robustness when dealing with uncommon scenarios.

Our contributions are summarized as follows:

1. We identify a dual-level reasoning process in the autoregressive image generation task by introducing the semantic-level and token-level CoT, which decouple high-level image planning from low-level pixel generation for more reliable generation.
2. We develop BiCoT-GRPO, a new reinforcement learning framework that jointly optimizes both levels of CoT reasoning, seamlessly integrating the understanding capabilities of ULMs for image generation. For reward modeling, we investigate a robust reward system utilizing an ensemble of vision experts.
3. Our resulting model, T2I-R1, incorporates both levels of CoT using BiCoT-GRPO and demonstrates significant quantitative and qualitative improvements, surpassing FLUX.1 across multiple established benchmarks.

## 2 Method

### 2.1 Preliminary

Recently, the employment of reinforcement learning has been the dominant approach to elicit the reasoning capability of the large models. [73] introduces GRPO, enhancing PPO by eliminating the value function and estimating the advantage in a group-relative manner. For a specific prompt-answer pair $(p, a)$, a group of $G$ individual responses $\{o_i\}_{i=1}^{G}$ is sampled from the old policy $\pi_{\theta_{\text{old}}}$. Each response is then input to a reward function to obtain the individual reward $\mathcal{R}_i$. Then, the advantage of the $i$-th response is calculated by normalizing the rewards $\{\mathcal{R}_i\}_{i=1}^{G}$ of the group:

$$A_i = \frac{\mathcal{R}_i - \text{mean}(\{\mathcal{R}_i\}_{i=1}^{G})}{\text{std}(\{\mathcal{R}_i\}_{i=1}^{G})}. \tag{1}$$

GRPO adopts a clipped objective similar to PPO. Besides, a KL penalty term between the current policy $\pi_\theta$ and the reference model $\pi_{\theta_{\text{ref}}}$ is directly added in the loss function:

$$\mathcal{J}_{\text{GRPO}}(\theta) = \mathbb{E}_{(q,a) \sim \mathcal{D}, \{o_i\}_{i=1}^{G} \sim \pi_{\theta_{\text{old}}}(\cdot|q)}$$
$$\left[ \frac{1}{\sum_{i=1}^{G} |o_i|} \sum_{i=1}^{G} \sum_{t=1}^{|o_i|} \left( \min\left(r_{i,t}(\theta)\hat{A}_i, \text{clip}\left(r_{i,t}(\theta), 1-\varepsilon, 1+\varepsilon\right)\hat{A}_i\right) - \beta D_{\text{KL}}(\pi_\theta||\pi_{\text{ref}}) \right) \right],$$

where $r_{i,j}(\theta)$ is the ratio between the probabilities of $\pi_\theta$ and $\pi_{\theta_{\text{old}}}$ for outputting the current token:

$$r_{i,j}(\theta) = \frac{\pi_\theta(o_{i,j} \mid q, o_{i,<j})}{\pi_{\theta_{\text{old}}}(o_{i,t} \mid q, o_{i,<j})}. \tag{2}$$

In text reasoning tasks like mathematical problem solving, the model is instructed to follow the pre-defined template to output the reasoning process and final answer. The reward functions are rule-based rewards that only check the correctness of the final answer and the output format.

### 2.2 Semantic-level and Token-level CoT

In the autoregressive text generation tasks of LLMs and LMMs, CoT occurs in the textual reasoning format. However, in autoregressive image generation tasks, we identify two distinct types of CoT that could enhance the image generation at different abstraction levels:

**Semantic-level CoT.** Semantic-level CoT is defined as the textual reasoning that precedes image generation, serving as an overall semantic planning stage for the intended image. This process mirrors human artistic creation: when given a brief prompt, an artist first thinks about the scene construction, considering object attributes, spatial relationships, and interactions. In addition to the planning for common prompts, we also observe the semantic-level CoT benefits two other scenarios. If the prompt does not directly depict the object to generate, the semantic-level CoT can reason about the true intention from the user's prompt, providing more aligned images. As illustrated in Fig. 2, the semantic-level CoT reasons that the flower cultivated in the country where Amsterdam is located is tulip. Without this semantic-level CoT, Janus-Pro fails to provide valid results. Additionally, the semantic-level CoT demonstrates importance when handling unusual or potentially ambiguous scenes. In the bottom example of Fig. 2, when given the prompt *'A pig on the bottom of a train'*, semantic-level CoT introduces the action 'lying' for the pig, creating a more sensible scenario. In contrast, direct generation without this interpretive imagination creates significant confusion for Janus-Pro. Formally, each semantic-level CoT $s_i$ is composed of $|s_i|$ text tokens $\{s_{i,1}, s_{i,2}, ..., s_{i,|s_i|}\}$.

**Token-level CoT.** Unique to the image generation task, a token-level step-by-step thinking exists in the image generation process. The generation of image tokens much resembles a chain of thought: the image tokens are generated patch by patch, where the current patch is generated based on the previous ones. We define the sequential generation of image tokens as token-level CoT. This process parallels how an artist progressively fills a canvas, with the generated patches forming a visual reasoning chain. The reasoning content is the choice of the specific visual token for each patch, which corresponds to the patch coherence, object appearance, lighting conditions, and other visual details. Note that,

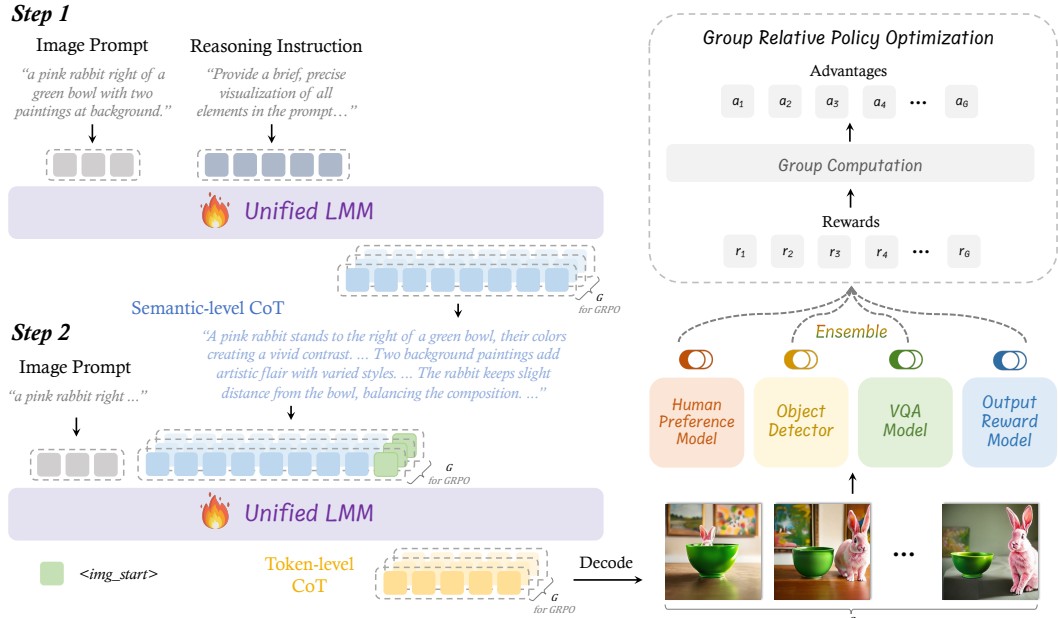

Figure 3: **Framework of BiCoT-GRPO.** In step 1, we instruct the model to generate the semantic-level CoT based on the image prompt. In step 2, images are generated conditioned on both the image prompt and semantic-level CoT, with the intermediate generation process serving as token-level CoT. The resulting images are evaluated by an ensemble of vision experts to obtain rewards. We generate $N$ images from each prompt to compute the group-relative reward and perform GRPO training.

the reasoning occurs simultaneously with final output generation, rather than as separate steps. The model reasons and generates in parallel, integrating its thinking directly into the creation process. The generated chain of patches is later reshaped to a 2D grid $G \in \mathbb{R}^{h \times w \times c}$ and input to an image decoder $\mathbb{D}$ to obtain the image. Unlike semantic-level CoT, which addresses global planning, token-level CoT focuses on local details and visual coherence across the image space. Formally, each token-level CoT $t_i$ consists of $M$ image tokens $\{t_{i,1}, t_{i,2}, ..., t_{i,M}\}$, where $M$ represents the resolution of the generated image, i.e., $M = h \times w$.

## 2.3 BiCoT-GRPO

GRPO has been proven to be highly effective for exploring the reasoning capability of the LLMs and LMMs. To accommodate both semantic-level and token-level CoT in image generation, we propose BiCoT-GRPO, where the model reasons twice in a single generation process. We instruct the model to first perform semantic-level CoT for global planning, and then dive into the local details by performing token-level CoT.

However, compared with the task of text generation, a great pipeline challenge is posed for incorporating two levels of CoT for image generation. Limited by the training paradigm, most current ULMs cannot generate interleaved images and text themselves. A manual signifier is often needed to instruct the model on which task to perform, either text generation or image generation. For Janus-Pro to generate an image, which is the ULM we use in this work, we need to manually concatenate an image start token (`<img_start>`) to explicitly instruct the model to start generating image tokens.

To tackle this problem, we propose a novel pipeline to facilitate ULM in generating images with two levels of CoT, as shown in Fig. 3. Specifically, our pipeline is composed of a two-step generation process. The first step is to generate the semantic-level CoT. We input the image prompt and instruct the model to imagine and reason about the details of the image to generate semantic-level CoT $\{s_i\}_{i=1}^G$. The second stage focuses on the token-level CoT generation. We input the image prompt, the generated semantic-level CoT in the first stage, and the image start token to the ULM for generating image tokens $\{t_i\}_{i=1}^G$. Then, the image tokens are input to the image decoder to obtain the

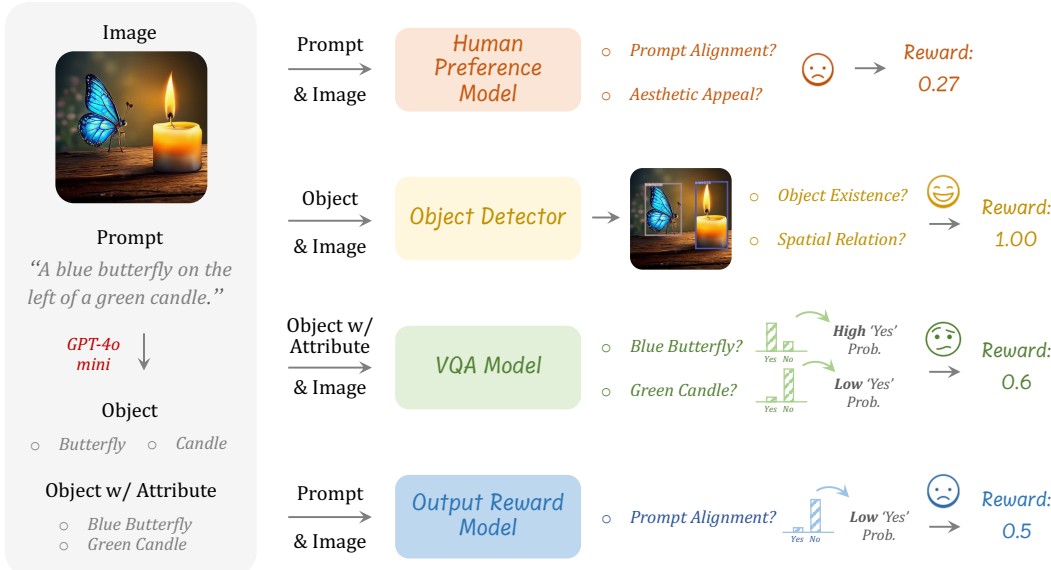

Figure 4: **Illustration of the Ensemble of Generation Rewards.** We use GPT-4o mini to extract the objects and their attributes before training. Each specialized reward model receives customized information inputs for the reward calculation. We take the average of all the rewards as final reward.

image $I$. Since there exist two types of CoT in our method, first the semantic-level CoT and then the token-level CoT. Each response $o_i$ is composed of two parts, namely $o_i = (s_i, t_i)$. In this sense, the $r_{i,j}(\theta)$ is converted to:

$$
r_{i,j}(\theta) = \frac{\pi_\theta(o_{i,j} \mid q, o_{i,<j})}{\pi_{\theta_{\text{old}}}(o_{i,j} \mid q, o_{i,<j})} = \begin{cases} \frac{\pi_\theta(s_{i,j}|q,s_{i,<j})}{\pi_{\theta_{\text{old}}}(s_{i,j}|q,s_{i,<j})}, & 0 \leq j \leq |s_i| \\ \frac{\pi_\theta(t_{i,j}|q,s_i,t_{i,<j})}{\pi_{\theta_{\text{old}}}(t_{i,j}|q,s_i,t_{i,<j})}, & |s_i| < j \leq |s_i| + M \end{cases} \tag{3}
$$

Then, we update the ULM by maximizing Equation 2.1. In practice, we incorporate the token-level policy gradient loss in [101], where the loss term is normalized over all the generated tokens to balance the reward on overly long semantic-level CoT.

## 2.4 Ensemble of Generation Rewards

Unlike DeepSeek-R1 with the rule-based reward, assessing the images based on pre-defined rules is infeasible. The assessment of the image includes various aspects, including the aesthetic appeal and objects' existence, attributes, and relationships. Considering the complexity, we introduce an ensemble of vision experts to judge the generated image from multiple aspects. Meanwhile, the use of multiple reward functions also serves as a regularization method to prevent the ULM from hacking into a specific reward model. As shown in Fig. 4, the ensemble contains the following experts:

**Human Preference Model.** Human preference models (HPMs), such as HPS [90] and ImageReward [94], are trained to simulate human aesthetic preferences. These models are developed using datasets of human rankings on synthetic images, where annotators evaluate and compare generated outputs. During inference, these models assess both the aesthetic quality and prompt alignment of a generated image, producing a composite human preference score $\mathcal{R}_{\text{HPM}}$. This expert provides a holistic reward signal from a general perspective.

**Object Detector.** Another option of the reward model is an object detector, e.g., GroundingDINO [49] and YOLO-world [12]. These open-vocabulary detection models accept an image along with object queries as input and output both the spatial positions and confidence scores for detected objects. This kind of vision expert serves as an ideal tool to evaluate the object's existence

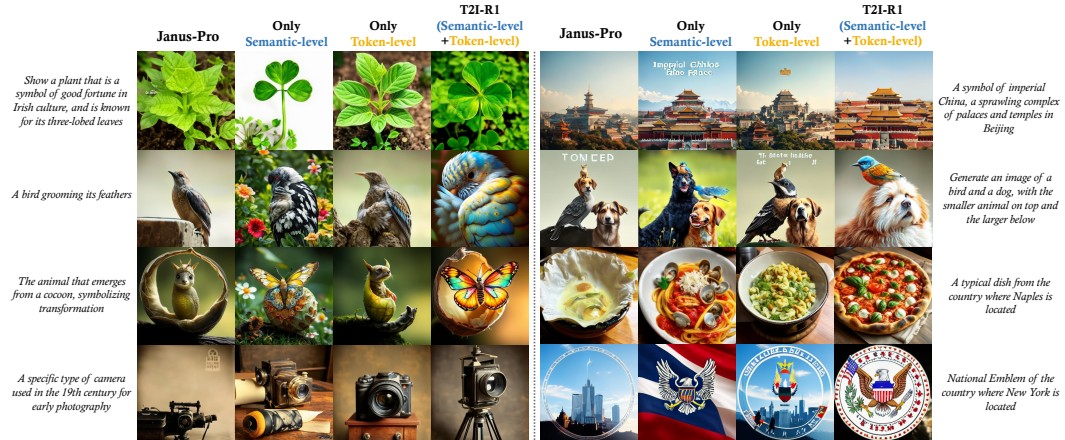

|  | Janus-Pro | Only Semantic-level | Only Token-level | T2I-R1 (Semantic-level +Token-level) | | Janus-Pro | Only Semantic-level | Only Token-level | T2I-R1 (Semantic-level +Token-level) |

*Show a plant that is a symbol of good fortune in Irish culture, and is known for its three-lobed leaves*

*A symbol of imperial China, a sprawling complex of palaces and temples in Beijing*

*A bird grooming its feathers*

*Generate an image of a bird and a dog, with the smaller animal on top and the larger below*

*The animal that emerges from a cocoon, symbolizing transformation*

*A typical dish from the country where Naples is located*

*A specific type of camera used in the 19th century for early photography*

*National Emblem of the country where New York is located*

Figure 5: **Visualization Results.** We provide the image generation results of the same prompt from four models: base model, the model with only semantic-level CoT optimized, the model with only token-level CoT optimized, and the model with both levels of CoT optimized.

and relationship concerning space and numbers. For implementation, we extract all objects $\{obj_i\}_{i=1}^{K}$ from the training image prompts, where $K$ represents the total number of objects. We then query the object detector to identify these objects within the generated image. For each object, we assign a binary existence score (1 if detected, 0 otherwise) and average these scores across all objects in the prompt. If the prompt contains a spatial relationship, we further leverage the detected location to validate its correctness. We calculate the relative distance and intersection over union (IoU) between the objects for the spatial score $\mathcal{R}_{\text{spatial}}$. If the number of the object $n_{obj_i}$ is specifically pointed out in the prompt, we compare the number with the detected number of the object $\hat{n}_{obj_i}$. The reward from the object detector $\mathcal{R}_{\text{Det}}$ is determined as:

$$
\mathcal{R}_{\text{Det}} = \begin{cases} \alpha \mathcal{R}_{\text{spatial}} + (1-\alpha)\frac{1}{K}\sum_{i=1}^{K}\mathbb{I}(obj_i \text{ detected}), & \text{if spatial relationship in the prompt,} \\ \frac{1}{n}\sum_{i=1}^{K}\mathbb{I}(n_{obj_i} = \hat{n}_{obj_i}), & \text{if number in the prompt,} \\ \frac{1}{n}\sum_{i=1}^{K}\mathbb{I}(obj_i \text{ detected}), & \text{else,} \end{cases}
$$

where $\mathcal{R}_{\text{spatial}}$ is 1 if the relative distance between the objects is larger than a threshold and the direction is right. If the direction is wrong, the reward is 0. Otherwise, we use the IoU as the spatial reward. We set $\alpha$ as 0.6 to encourage the correctness of the spatial relationship.

**Visual Question Answering Model.** The visual question answering (VQA) models are trained to answer questions based on the image input. The VQA models include earlier models prior to LLM, e.g., BLIP [42] and GIT [82], and LMMs like LLaVA [47]. We leverage these models to judge the existence and attributes of the objects. For example, if the image prompt is *a red dog and a yellow cat*, we first reformat each individual object with its attribute as a question to the VQA model, i.e., *a red dog?* and *a yellow cat?*. Then, we record the probability for the model to answer *Yes* as $P_{\text{Yes}}^{i}$ and *No* as $P_{\text{No}}^{i}$. The reward for a prompt is calculated as: $\mathcal{R}_{\text{VQA}} = \frac{1}{K}\sum_{i}\frac{P_{\text{Yes}}^{i}}{P_{\text{Yes}}^{i}+P_{\text{No}}^{i}}$.

**Output Reward Model.** Lastly, we also employ the output reward model (ORM) proposed in [24] as a reward model. The ORM is fine-tuned from an LMM (e.g., LLaVA-OneVision [39]) specifically for evaluating the alignment between the prompt and the image. The fine-tuning is to instruct the model to output *Yes* if the image perfectly aligns with the image and *No* otherwise. We calculate $\mathcal{R}_{\text{ORM}}$ using the methodology similar to $\mathcal{R}_{\text{VQA}}$, except that we input the whole image prompt to the ORM instead of reformatting the prompt. The major difference between the ORM and HPMs is that the ORM model incorporates extensive world knowledge inside the LMM while HPMs mainly focus on the human preferences including prompt-image alignment and aesthetic appeal.

We can choose one or multiple reward functions illustrated above, and take the average as the final reward for a specific sample. The detailed experiments of reward model are shown in Table 3.

Table 1: **T2I-CompBench Result.** The best score is in blue , with the second-best score in green .

| Model | Attribute Binding | | | Object Relationship | | Complex↑ |
|---|---|---|---|---|---|---|
| | Color ↑ | Shape↑ | Texture↑ | Spatial↑ | Non-Spatial↑ | |
| *Diffusion Models* | | | | | | |
| SD-v1.5 [70] | 0.3758 | 0.3713 | 0.4186 | 0.1165 | 0.3112 | 0.3047 |
| PixArt-$\alpha$ [6] | 0.6690 | 0.4927 | 0.6477 | 0.2064 | 0.3197 | 0.3433 |
| CoMat [31] | 0.7827 | 0.5329 | 0.6468 | 0.2428 | 0.3187 | 0.3680 |
| SD-XL-base-1.0 [66] | 0.5879 | 0.4687 | 0.5299 | 0.2131 | 0.3119 | 0.3237 |
| FLUX.1 [37] | 0.7407 | 0.5718 | 0.6922 | 0.2863 | 0.3127 | 0.3703 |
| *AutoRegressive Models* | | | | | | |
| Show-o [93] | 0.5623 | 0.4178 | 0.4641 | 0.2015 | 0.3067 | 0.2992 |
| Show-o + PARM [24] | 0.7549 | 0.5632 | 0.6684 | 0.2971 | 0.3126 | 0.3701 |
| EMU3 [83] | 0.7544 | 0.5706 | 0.7164 | - | - | - |
| Janus-Pro-7B [9] (Baseline) | 0.6359 | 0.3528 | 0.4936 | 0.2061 | 0.3085 | 0.3559 |
| **T2I-R1 (Ours)** | 0.8130 | 0.5852 | 0.7243 | 0.3378 | 0.3090 | 0.3993 |

# 3 Experiment

In this section, we first provide the main results of T2I-R1 in T2I-CompBench [28], WISE [61] and GenAI-Bench [45] in Section 3.1. Then we present the results of different reward function combinations in Section 3.2 and the ablation study of the effectiveness of two levels of CoT in Section 3.3. Please refer to the Appendix B for TIIF-Bench [85] results, detailed experiment setup, and more visualizations.

## 3.1 Main Results

We compare T2I-R1 with leading text-to-image diffusion and autoregressive models on the T2I-CompBench and WISE benchmarks (in Table 1, 2 and 4). We also provide the qualitative results in Fig. 5. Our method demonstrates substantial improvements over the baseline model, with average enhancements of 13% and 19% on T2I-CompBench and WISE, respectively. On T2I-CompBench, the most significant gains appear in attribute binding, with an average improvement of 19%. For the WISE benchmark, improvements are more evenly distributed across categories. When compared to the more powerful state-of-the-art diffusion models, T2I-R1 achieves superior or comparable results across both benchmarks. Notably, on T2I-CompBench, our method leads in five of six subtasks, with an exceptional performance in the spatial subtask (0.3378), surpassing previous SOTA results by over 5%. Similarly, for WISE, T2I-R1 excels in four of seven subtasks and achieves the highest overall score of 0.54, outperforming the robust FLUX.1-dev by 4%. Remarkably, our approach consistently achieves the leading results across all subtasks in both benchmarks when compared to other autoregressive models. Remarkably, the improvement on T2I-Compbench benefits from the planning ability brought by the semantic-level CoT, which designs the complex scenarios before generation. While the enhancement of WISE is due to the reasoning capability from the semantic-level CoT, which deduces the true object or place depicted behind the prompt. For GenAI-Bench, T2I-R1 largely improves the baseline model, and in the meantime, achieves the highest overall score on both the basic and advanced prompts. Again, T2I-R1 surpasses FLUX.1 [37] in both types of prompts and showcases a remarkable margin in the advanced prompt, probably attributed to the high-level reasoning capability granted by semantic-level CoT.

## 3.2 Reward Analysis

In this section, we experiment with the choice of reward functions and their combinations. We hope to provide some insights into how to choose the reward functions and combine them. Our results are shown in Table 3. We first experiment with the individual reward model. HPM (H) demonstrates superior performance in attribute binding but shows limited effectiveness in object relationships, likely due to its weak relation comprehension capabilities. The object detector (D) yields the least improvement in attribute binding, which aligns with expectations since our detector-based reward functions do not explicitly evaluate attributes. The improvements observed stem solely from enhanced object existence ratios in the prompts. We observe that VQA model (V) and ORM (O) are

Table 2: **WISE Result.** The best score is in blue , with the second-best score in green .

| Model | Cultural↑ | Spatio-Temporal | | Natural Science | | | Overall |
|---|---|---|---|---|---|---|---|
| | | Time↑ | Space↑ | Biology ↑ | Physics↑ | Chemistry↑ | |
| *Diffusion Models* | | | | | | | |
| PixArt-Alpha [6] | 0.45 | 0.50 | 0.48 | 0.49 | 0.56 | 0.34 | 0.47 |
| Playground-v2.5 [40] | 0.49 | 0.58 | 0.55 | 0.43 | 0.48 | 0.33 | 0.49 |
| SD-v1-5 [70] | 0.34 | 0.35 | 0.32 | 0.28 | 0.29 | 0.21 | 0.32 |
| SD-XL-base-0.9 [66] | 0.43 | 0.48 | 0.47 | 0.44 | 0.45 | 0.27 | 0.43 |
| FLUX.1-dev [37] | 0.48 | 0.58 | 0.62 | 0.42 | 0.51 | 0.35 | 0.50 |
| *AutoRegressive Models* | | | | | | | |
| Emu3 [83] | 0.34 | 0.45 | 0.48 | 0.41 | 0.45 | 0.27 | 0.39 |
| Show-o [93] | 0.28 | 0.40 | 0.48 | 0.30 | 0.46 | 0.30 | 0.35 |
| VILA-U [91] | 0.26 | 0.33 | 0.37 | 0.35 | 0.39 | 0.23 | 0.31 |
| Janus-Pro-7B [9] (Baseline) | 0.30 | 0.37 | 0.49 | 0.36 | 0.42 | 0.26 | 0.35 |
| **T2I-R1 (Ours)** | 0.56 | 0.55 | 0.63 | 0.54 | 0.55 | 0.30 | 0.54 |

Table 3: **T2I-CompBench Results with Different Reward Models.** 'Det' stands for object detector.

| Model | Reward Model | | | | Attribute Binding | | | Object Relationship | | Complex↑ | Visual Quality↑ |
|---|---|---|---|---|---|---|---|---|---|---|---|
| | HPM | Det | VQA | ORM | Color ↑ | Shape↑ | Texture↑ | Spatial↑ | Non-Spatial↑ | | |
| Janus-Pro-7B | - | - | - | - | 0.6359 | 0.3528 | 0.4936 | 0.2061 | 0.3085 | 0.3559 | - |
| - | ✓ | - | - | - | 0.8134 | 0.6048 | 0.7311 | 0.2383 | 0.3012 | 0.3899 | - |
| - | - | ✓ | - | - | 0.7422 | 0.5140 | 0.6494 | 0.3044 | 0.3100 | 0.3872 | - |
| - | - | - | ✓ | - | 0.8171 | 0.6019 | 0.7307 | 0.2969 | 0.3088 | 0.4052 | 0.218 |
| - | - | - | - | ✓ | 0.7819 | 0.5638 | 0.7010 | 0.3301 | 0.3103 | 0.3959 | 1.775 |
| - | ✓ | ✓ | - | - | 0.8210 | 0.6074 | 0.7440 | 0.3189 | 0.3076 | 0.4005 | 1.942 |
| **T2I-R1** | ✓ | ✓ | ✓ | - | 0.8130 | 0.5852 | 0.7243 | 0.3378 | 0.3090 | 0.3993 | 2.063 |
| - | ✓ | ✓ | ✓ | ✓ | 0.7599 | 0.5742 | 0.6902 | 0.2796 | 0.3070 | 0.3921 | - |

both effective reward models with distinct strengths: the VQA model excels at improving attribute binding, while ORM demonstrates superior performance in relationships. Then we experiment with multiple reward models. We start from the composition of HPM and object detector (H + D), and progressively incorporate other reward models. Our findings indicate that both the HPM-object detector combination (H + D) and the three-model integration of HPM, object detector, and VQA (H + D + V) deliver balanced and satisfactory results in both attribute and relationship tasks. To obtain the optimal choice of reward models, we conduct a human study to evaluate the visual quality, detailed in Appendix C.2. We adopt the combination of the highest visual quality, the ensemble of three reward models (H + D + V) for our final model.

Token-level CoT Only          Semantic-level + Token-level CoT

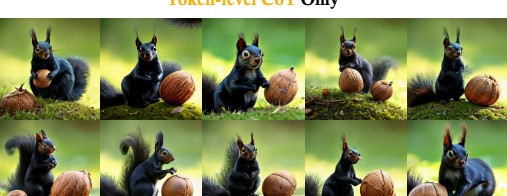 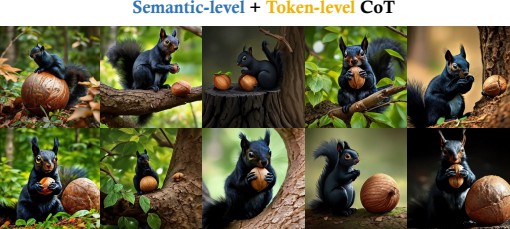

*a black squirrel and a brown nut*

Figure 6: **Visualization Result of the Image Diversity of a Single Prompt.** We showcase the result of only token-level CoT optimized and both semantic-level and token-level CoT optimized.

## 3.3 Ablation Study

To validate the effectiveness of the semantic-level CoT, we compare T2I-R1 with a baseline method that generates images using only the token-level CoT optimized with the GRPO method. This is the default text-to-image generation setting in Janus, whose result is shown in the third row in Table 5. Comparing the third and fourth row in the table, we find that semantic-level CoT generally brings performance improvements across both benchmarks tested. We witness a particularly significant gain on the WISE benchmark. This enhanced performance can be attributed to the textual reasoning

Table 4: **GenAI-Bench Evaluation Results.** The best score is in `blue`, with the second-best score in `green`.

| Method | Basic Prompt | | | | | | Advanced Prompt | | | | | |
| --- | --- | --- | --- | --- | --- | --- | --- | --- | --- | --- | --- | --- |
| | Attribute↑ | Scene↑ | Relation | | | Overall↑ | Count↑ | Differ↑ | Compare↑ | Logical | | Overall↑ |
| | | | Spatial↑ | Action↑ | Part↑ | | | | | Negate↑ | Universal↑ | |
| *Diffusion Models* | | | | | | | | | | | | |
| SD v2.1 [70] | 0.80 | 0.79 | 0.76 | 0.77 | 0.80 | 0.78 | 0.68 | 0.70 | 0.68 | 0.54 | 0.64 | 0.62 |
| SD-XL [66] | 0.84 | 0.84 | 0.82 | 0.83 | 0.89 | 0.83 | 0.71 | 0.73 | 0.69 | 0.50 | 0.66 | 0.63 |
| Midjourney v6 [60] | 0.88 | 0.87 | 0.87 | 0.87 | 0.91 | 0.87 | 0.78 | 0.78 | 0.79 | 0.50 | 0.76 | 0.69 |
| FLUX.1-dev [37] | 0.87 | 0.88 | 0.87 | 0.85 | 0.87 | 0.87 | 0.75 | 0.78 | 0.74 | 0.45 | 0.70 | 0.64 |
| *Auto-Regressive Models* | | | | | | | | | | | | |
| LWM [46] | 0.63 | 0.62 | 0.65 | 0.63 | 0.70 | 0.63 | 0.59 | 0.58 | 0.54 | 0.49 | 0.52 | 0.53 |
| Show-o [93] | 0.72 | 0.72 | 0.70 | 0.70 | 0.75 | 0.70 | 0.70 | 0.62 | 0.71 | 0.51 | 0.65 | 0.60 |
| VILA-U [91] | 0.78 | 0.78 | 0.77 | 0.78 | 0.79 | 0.76 | 0.70 | 0.71 | 0.74 | 0.53 | 0.66 | 0.64 |
| Liquid [87] | – | – | – | – | – | – | 0.76 | 0.73 | 0.74 | 0.46 | 0.74 | 0.65 |
| UniTok [56] | – | – | – | – | – | – | 0.76 | 0.76 | 0.79 | 0.46 | 0.73 | 0.67 |
| Mogao-7B [44] | – | – | – | – | – | – | 0.77 | 0.74 | 0.77 | 0.53 | 0.71 | 0.68 |
| Janus-Pro-7B [9] (Baseline) | 0.85 | 0.87 | 0.85 | 0.84 | 0.85 | 0.84 | 0.73 | 0.73 | 0.71 | 0.48 | 0.65 | 0.65 |
| **T2I-R1 (Ours)** | 0.87 | 0.89 | 0.89 | 0.87 | 0.87 | 0.88 | 0.81 | 0.82 | 0.78 | 0.60 | 0.73 | 0.73 |

Table 5: **Ablation Experiments on the Effectiveness of the Two Levels of CoT.**

| Model | Optimized CoT | | T2I-CompBench | | | WISE | | | Diversity↑ |
| --- | --- | --- | --- | --- | --- | --- | --- | --- | --- |
| | Semantic-level | Token-level | Color↑ | Shape↑ | Texture↑ | Culture↑ | Spatio-Temporal↑ | Science↑ | |
| Janus-Pro-7B | | | 0.6359 | 0.3528 | 0.4936 | 0.3000 | 0.4232 | 0.3467 | 6.976 |
| - | ✓ | | 0.8082 | 0.5684 | 0.7219 | 0.4900 | 0.5599 | 0.4367 | 8.177 |
| - | | ✓ | 0.7752 | 0.5849 | 0.7451 | 0.3500 | 0.4732 | 0.3900 | 6.255 |
| **T2I-R1** | ✓ | ✓ | 0.8130 | 0.5852 | 0.7243 | 0.5600 | 0.5855 | 0.4633 | 8.203 |

capabilities inherent in semantic-level CoT. As illustrated in Fig. 5, our method could first clearly reason about the objects or phenomena described in the prompt through semantic-level CoT. This effectively decouples the reasoning and generation processes and thereby facilitates superior results. We also observe that training solely with token-level CoT substantially reduces the diversity of generated images, as demonstrated in Fig. 6, 7, 13, and 14. To quantify this effect, we evaluate image diversity by reusing the generated images from T2I-CompBench, where each prompt generates ten images. We compute the Vendi Score [18] across the ten images for each prompt. Results indicate that GRPO training without semantic-level CoT decreases the diversity score, whereas incorporating semantic-level CoT significantly improves diversity through varied textual planning.

We also consider another situation to validate the effectiveness of token-level CoT: the semantic-level CoT is incorporated in the image generation process, as T2I-R1, but GRPO only optimizes the semantic-level CoT without the token-level CoT. This can be viewed as only enhancing the model's high-level planning capabilities. The second row of Table 5 presents the result. The results show that optimizing semantic-level CoT exclusively yields smaller improvements compared to the joint optimization approach. Additionally, we find that optimizing both CoT types produces images with much better aesthetic quality compared with optimizing semantic-level CoT only, as shown in Fig. 5. This indicates the necessity to jointly optimize both levels of CoT.

## 4 Conclusion

In this paper, we introduce T2I-R1, the first reasoning-enhanced text-to-image model powered by a bi-level CoT reasoning process. We identify the semantic-level CoT for high-level planning and the token-level CoT for patch-by-patch generation. We further integrate them through our proposed BiCoT-GRPO, an RL framework incorporating two levels of CoT within the same training step. By leveraging a ULM capable of both visual understanding and generation, our approach eliminates the need for separate specialized models while achieving significant performance improvements, +13% on T2I-CompBench and +19% on the WISE benchmark. Our qualitative analysis demonstrates that T2I-R1 better understands complex prompts, reasons about user intentions, and handles uncommon scenarios with greater robustness, establishing a new paradigm for text-to-image generation tasks.

## Acknowledgments

This study was supported in part by National Key R&D Program of China Project 2022ZD0161100, in part by the Centre for Perceptual and Interactive Intelligence, a CUHK-led InnoCentre under the InnoHK initiative of the Innovation and Technology Commission of the Hong Kong Special Administrative Region Government, in part by NSFC-RGC Project N_CUHK498/24, and in part by Guangdong Basic and Applied Basic Research Foundation (No. 2023B1515130008, XW).

Additional support was provided by the Research Grants Council of the Hong Kong Special Administrative Region, China, under Project 14201321 and Project 14200824.

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

# A Related Work

**Unified Generation and Understanding LMM.** Recently, the effort to unify image generation and understanding in a single LMM has attracted much attention. Building upon large language models (LLMs), it is natural for the LMMs to understand the image and output the text [63, 39, 109, 19, 104, 33]. However, the method of how to generate an image from a LMM is still under exploration. The image generation method diverges into different branches. One line of the method relies on an exterior image generation model to complete generation [15, 78, 77, 43, 80, 17, 108, 38]. The generator often utilizes text-to-image diffusion models [70, 66] due to its powerful generation capability. To deliver the generation information, the LMM passes either the implicit conditional feature or the explicit image prompt to the generator. For example, EMU [78] first trains the LMM to output CLIP [68] image features identical to that input to the LMM. Then, a pretrained UNet [71] of Stable Diffusion [70] receives the output feature as the condition to generate an image. Another line of the method seeks to train the LMM to generate discrete tokens produced by VQGAN [16] to eliminate the need for an additional generator. [83, 41] directly adopts the VQGAN encoder as the image tokenizer for LMM. However, the VQGAN encoder is only pretrained on the image reconstruction task and thereby generates visual tokens less helpful for image understanding. To improve the understanding capability, [86, 9, 57, 48] proposes to tackle the understanding and generation tasks with different vision encoders separately. The CLIP encoder deals with image input for understanding, while the VQGAN encoder is responsible for generation. Moreover, some works [91, 67, 75] attempt to empower the vision encoder with both the understanding and the generation capability. VILA-U [91] trains a vision encoder with both the contrastive loss [68] for text-image understanding and reconstruction loss [16] for image detail preserving. Thanks to the joint pretraining, the vision encoder could generate text-aligned discrete visual tokens. The LMM is then trained to receive the discrete tokens for image understanding and predict them for image generation.

**Reinforcement Learning for Large Reasoning Models.** The emergence of OpenAI o1 [65] has gained tremendous attention in developing the reasoning capability of large language models. Later, DeepSeek-R1 [20] proposes a rule-based reward and GRPO training method. The introduced method instructs the model to perform an extensive reasoning process before generating the final answer. The reward only focuses on the correctness of the final answer and the following of the pre-defined format. Recently, a number of works have applied this method to multi-modal large language models [7, 59, 97, 103, 14, 29, 11] with task-specific rewards like correctness and IoU [50]. This training paradigm largely helps various reasoning-intensive tasks [69, 32, 23] like mathematical problem-solving [27, 58, 51, 105, 106], code generation [8, 2, 30], and complex scene understanding [98].

# B More Experiment Details

## B.1 Experiment Setup

**Training Settings.** Our training dataset comprises text prompts sourced from the training set of T2I-CompBench [28] and [24], totaling 6,786 prompts with no images. Prior to training, we use GPT-4o mini to extract the objects and their attributes from the prompts to facilitate computing the rewards. We use Janus-Pro-7B as the base model. We use a learning rate of 1e-6 and a beta of 0.01. For the reward model, we choose HPS [90] as the human preference model, GroundingDINO [49] as the object detector, and GIT [82] as the VQA model. For the ORM, we finetune LLaVA-OneVision-7B in the same manner as [24].

**Benchmark.** We test on T2I-CompBench [28], WISE [61], GenAI-Bench [45], and TIIF-Bench [85] to validate the effectiveness of our method. T2I-CompBench comprises 6,000 compositional text prompts evaluating three categories (attribute binding, object relationships, and complex compositions) and six sub-categories (color binding, shape binding, texture binding, spatial relationships, non-spatial relationships, and complex compositions). WISE consists of 1,000 text prompts spanning three categories (cultural common sense, spatial-temporal reasoning, and natural science) for evaluating world knowledge of the text-to-image models. To correctly generate an image, the model needs to reason about what the exact object or scenario is depicted in the prompt. We slightly modify the reasoning instruction on the WISE benchmark for more aligned results. GenAI-

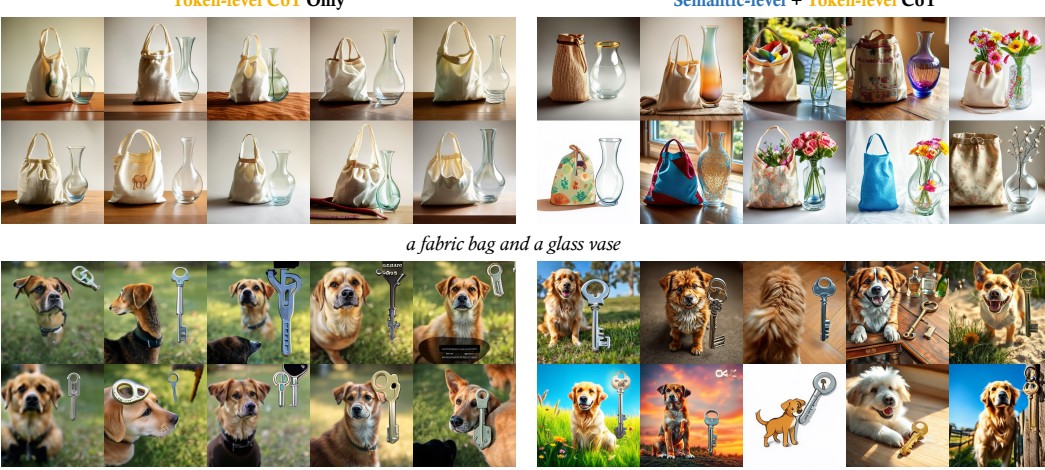

| Token-level CoT Only | Semantic-level + Token-level CoT |

*a fabric bag and a glass vase*

*a key on the right of a dog*

Figure 7: **More Visualization Result of the Image Diversity of a Single Prompt.** We showcase the result of only token-level CoT optimized and both semantic-level and token-level CoT optimized.

Table 6: **TIIF-Bench Testmini Subset Evaluation Results.** The best score is in blue, with the second-best score in green.

| Model | Overall | | Basic Following | | | | | | | | Advanced Following | | | | | | | | | | | | | | Designer | |
|---|---|---|---|---|---|---|---|---|---|---|---|---|---|---|---|---|---|---|---|---|---|---|---|---|---|---|
| | | | Avg | | Attribute | | Relation | | Reasoning | | Avg | | Attribute +Relation | | Attribute +Reasoning | | Relation +Reasoning | | Style | | Text | | Real World | |
| | short | long | short | long | short | long | short | long | short | long | short | long | short | long | short | long | short | long | short | long | short | long | short | long |
| Llamagen [76] | 41.67 | 38.22 | 53.00 | 50.00 | 48.33 | 42.33 | 59.57 | 60.32 | 51.07 | 47.32 | 35.89 | 32.61 | 38.82 | 31.57 | 40.84 | 47.22 | 49.59 | 46.22 | 46.67 | 33.33 | 0.00 | 0.00 | 39.73 | 35.62 |
| LightGen [89] | 53.22 | 43.41 | 66.58 | 47.91 | 55.83 | 47.33 | 74.82 | 45.82 | 69.07 | 50.57 | 46.74 | 41.53 | 62.44 | 40.82 | 61.71 | 50.47 | 50.34 | 45.34 | 53.33 | 53.33 | 0.00 | 6.83 | 50.92 | 50.55 |
| Show-o [93] | 59.72 | 58.86 | 73.08 | 75.83 | 74.83 | 79.83 | 78.82 | 78.32 | 65.57 | 69.32 | 53.67 | 50.38 | 60.95 | 56.82 | 68.59 | 68.96 | 66.46 | 56.22 | 63.33 | 66.67 | 3.83 | 2.83 | 55.02 | 50.92 |
| Infinity [25] | 62.07 | 62.32 | 73.08 | 75.41 | 74.33 | 76.83 | 72.82 | 77.57 | 72.07 | 71.82 | 56.64 | 54.98 | 60.44 | 55.57 | 74.22 | 64.71 | 60.22 | 59.71 | 80.00 | 73.33 | 10.83 | 23.83 | 54.28 | 56.89 |
| Janus-Pro [9] | 66.50 | 65.02 | 79.33 | 78.25 | 79.33 | 82.33 | 78.32 | 73.32 | 80.32 | 79.07 | 59.71 | 58.82 | 66.07 | 56.20 | 70.46 | 70.84 | 67.22 | 59.97 | 60.00 | 70.00 | 28.83 | 33.83 | 65.84 | 60.25 |
| **T2I-R1 (Ours)** | 68.59 | 67.19 | 82.90 | 81.63 | 86.50 | 83.00 | 83.47 | 79.43 | 78.73 | 82.46 | 69.05 | 68.00 | 71.64 | 69.47 | 72.43 | 69.95 | 69.40 | 70.40 | 60.00 | 63.33 | 27.60 | 26.24 | 67.54 | 60.45 |

Bench is a benchmark containing 1,600 complex, real-world text prompts collected from professional designers, which covers a broad spectrum of compositional text-to-visual generation elements, from basic aspects like scenes, attributes, and relationships to more professional ones, including counting, comparison, differentiation, and logical reasoning. TIIF-Bench is a comprehensive benchmark for fine-grained text-to-image model evaluation, featuring 36 novel prompt combinations across six compositional dimensions and 100 real-world designer-level prompts with rich aesthetic judgment. We follow the official evaluation setting of all the benchmarks.

# C   More Experiment Results

## C.1   More Results

We provide the experiment results on TIIF-Bench in Table 6 and more qualitative examples in Fig. 8.

Finally, we discuss the zero-shot potential of the baseline model to perform both semantic-level and token-level reasoning. Specifically, we apply the same image generation process of T2I-R1 directly to the baseline model, where the baseline model is first instructed to output the semantic-level CoT and then the token-level CoT. We term this method of generation as 'Janus-Pro w/ zero-shot semantic-level CoT' in Figure 9-12. As shown in the figure, zero-shot semantic-level CoT brings very marginal improvement, while T2I-R1 demonstrates a satisfying result. The reasons are twofold: (1) Zero-shot semantic-level CoT misses critical objects in the original prompt. As shown in Figure 12, the zero-shot semantic-level CoT misses the *bird* in the original prompt. (2) Zero-shot semantic-level CoT does not fit the model's generation ability or provide useful information for generation. Although

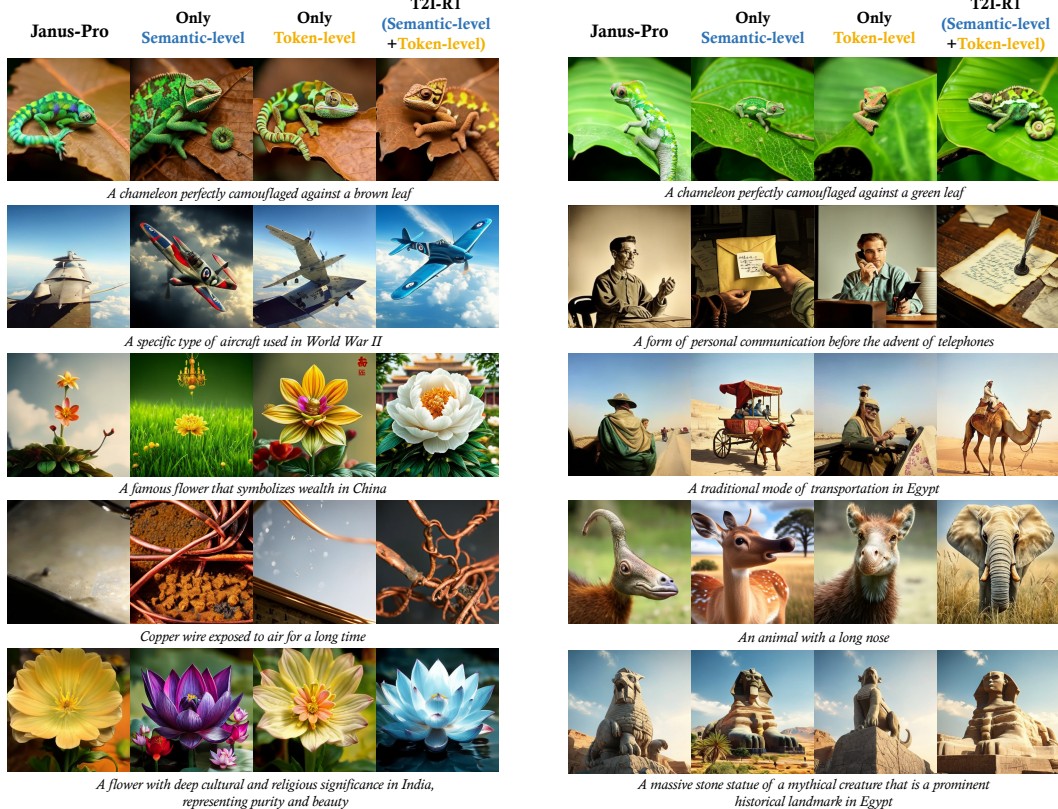

| Janus-Pro | Only Semantic-level | Only Token-level | T2I-R1 (Semantic-level +Token-level) | Janus-Pro | Only Semantic-level | Only Token-level | T2I-R1 (Semantic-level +Token-level) |

*A chameleon perfectly camouflaged against a brown leaf*

*A chameleon perfectly camouflaged against a green leaf*

*A specific type of aircraft used in World War II*

*A form of personal communication before the advent of telephones*

*A famous flower that symbolizes wealth in China*

*A traditional mode of transportation in Egypt*

*Copper wire exposed to air for a long time*

*An animal with a long nose*

*A flower with deep cultural and religious significance in India, representing purity and beauty*

*A massive stone statue of a mythical creature that is a prominent historical landmark in Egypt*

Figure 8: **More Visualization Results.** We provide the image generation results of the same prompt from four models: base model, the model with only semantic-level CoT optimized, the model with only token-level CoT optimized, and the model with both levels of CoT optimized.

the semantic-level CoT in Figure 9-11 includes all the objects and relationships, the baseline model still fails to generate a satisfying result. This highlights the necessity of our proposed BiCoT-GRPO training method to build the synergy between the two levels of CoT and make them work together.

## C.2  More Details about Reward Analysis

We conduct a human study to evaluate the visual quality of the generated images. Specifically, we select four options of reward models (V, O, H + D, and H + D + V) to generate an image from the same prompt. Then we ask humans to rank the four images and score them according to the rank (rank 1 for 3 points, rank 2 for 2 points, and so on). We ask the annotators to judge only according to the visual appeal. We provide examples of an unsatisfactory generation, including different aspects: 1) Corrupted floating words, 2) Distortion like melting effect of faces, hands, or other objects, 3) Extra or missing hands, legs, tails, etc., 4) Object merging, frequently observed for compositional prompts, 5) Low-level aspects including over exposure or over saturation, 6) Blurred areas, especially in fine details like keyboard. The instruction to the annotator highlights that the highly ranked images should show no or fewer problems compared with others. Eight graduate students are employed to conduct the study to eliminate individual bias. We randomly choose 30 prompts from each of the subtasks from the T2I-CompBench. The result is shown in the visual quality column in Table 3. We observe that ensemble rewards achieve better visual quality, with H + D + V obtaining slightly superior results. This improvement could be attributed to the implicit regularization provided by multiple rewards, preventing overfitting to a single reward model. Conversely, individual reward models fail to provide satisfactory quality despite high benchmark scores.

## C.3 Comparison with Prompt Rewriting Methods

The prompt rewriting [96, 102, 26, 88] is a method to leverage an external LLM to generate an enriched version of the prompt and use this version for the text-to-image model to produce the images. Our semantic-level CoT design resembles the high-level idea of this method, but there exist several key differences. First, the semantic-level CoT is generated from the image generation model itself, while the prompt enrichment leverages an extra LLM. Second, our design facilitates a joint-optimization of the prompt design and the image generation. On the contrary, the prompt enrichment is generation-model-agnostic, which means that although the prompt is enriched with more details, it is not necessary that the generation model can generate a better image based on this enriched prompt. We validate this claim in Figure 9-11. Although the zero-shot semantic-level CoT (enriched prompt) all correctly includes all the key objects mentioned in the prompt and adds more details, the model still cannot generate satisfying result. Simply enriching the prompt is not sufficient to bring notable improvements.

## C.4 Comparison with Training with RL and Supervised Finetuning

In our experiments, we directly employ reinforcement learning. An alternative is to first conduct supervised finetuning (SFT) and then continue with RL training. Our findings show that a cold start stage with SFT seems to be detrimental for the final performance. Specifically, we use the open-source high-quality text-to-image dataset BLIP3o-60K [5] as our SFT dataset. We employ Qwen2.5-VL-72B-Instruct [3] to generate the semantic-level CoT with the image and the original short prompt input. We finetune the model for one epoch and then conduct RL finetuning following the same training setting of T2I-R1. The results are shown in Table 7 below:

Table 7: **Model Performance Comparison Between RL and SFT.**

| Training Method | Color | Shape | Texture | Spatial | Non-Spatial | Complex |
|---|---|---|---|---|---|---|
| Janus-Pro | 0.6359 | 0.3528 | 0.4926 | 0.2061 | 0.3085 | 0.3559 |
| SFT Only | 0.7035 | 0.5217 | 0.6423 | 0.2775 | 0.3068 | 0.3626 |
| Hybrid optimization | 0.7765 | 0.5832 | 0.6981 | 0.3327 | 0.3092 | 0.3949 |
| T2I-R1 (RL Only) | 0.8130 | 0.5852 | 0.7243 | 0.3378 | 0.3090 | 0.3993 |

The key findings are two-fold. First, high-quality SFT is beneficial. The curated BLIP3-60k dataset yields significant performance gains. However, SFT performance substantially lags behind RL training. We hypothesize this occurs because SFT constrains the model to replicate the training distribution rather than leveraging its inherent capabilities. Specifically, when presented with a valid semantic-level CoT, SFT forces the model to generate the exact corresponding training image, even when the model could produce alternative valid outputs. This constraint introduces unnecessary training complexity. Second, RL after high-quality SFT is still inferior to direct RL training. While high-quality SFT pre-training improves subsequent RL performance, the combined approach remains mostly inferior to or merely comparable with direct RL training.

## C.5 Choice of Reward Weights

In Table 1, the weight of all the rewards are set to 1. We find that the weight of the reward model has little influence on the final training result. Here we provide the detailed study to illustrate the our method is stable to the weight of the reward model. We follow the training setting of T2I-R1, where three reward models, HPS, GroundingDINO, and GIT are employed. In our experiments, we multiply the weight of each reward model by 5 respectively, while maintaining the weight of the other two reward models. Apart from these three experiments, we also conduct an experiment where we compute the relative reward of each reward model inside the group, and then sum up the normalized reward as the final reward for the sample. This eliminates the mean and variance difference among the reward models (termed as Normalized Reward). The results are shown in Table 8 below.

## C.6 Hyperparameters

All of our experiments are conducted on 8 H800. Our training procedure lasts about 16 hours. We provide the detailed training hyperparameters in Table 9.

Table 8: **Comparison of Different Reward Weights.**

| Reward Design | Color | Shape | Texture | Spatial | Non-Spatial | Complex |
|---|---|---|---|---|---|---|
| 5∗HPS, 1∗others | 0.8215 | 0.5915 | 0.7337 | 0.3051 | 0.3074 | 0.4051 |
| 5∗GDINO, 1∗others | 0.7951 | 0.5520 | 0.7100 | 0.3313 | 0.3104 | 0.3916 |
| 5∗GIT, 1∗others | 0.7972 | 0.5620 | 0.7149 | 0.3357 | 0.3111 | 0.3934 |
| Normalized Reward | 0.8106 | 0.5820 | 0.7142 | 0.2940 | 0.3072 | 0.3993 |
| T2I-R1 (Equal Weights) | 0.8130 | 0.5852 | 0.7243 | 0.3378 | 0.3090 | 0.3993 |

Table 9: T2I-R1 training hyperparameters.

| Name | |
|---|---|
| Learning rate | 1e-6 |
| Beta $\beta$ | 0.01 |
| Group Size $G$ | 8 |
| Classifier-Free Guidance Scale | 5 |
| Max Gradient Norm | 1.0 |
| Batchsize | 8 |
| Training Steps | 1,600 |
| Gradient Accumulation Steps | 2 |
| Image Resolution $h \times w$ | $384 \times 384$ |

# D  Limitations and Future Work

While this work explores the text-to-image generation task, it requires more exploration on how to apply this paradigm to other modalities like video generation [4] or 3D [21] tasks. Specifically, video generation tasks are more complex regarding the reward design and the base model. For the reward design, how to apply dense rewards on each generated frame is still an open question. Besides, there exists no understanding and generation unified model for videos, so BiCoT-GRPO cannot be used directly. Meanwhile, the current inference time of video generation is too long for the current GRPO paradigm. How to balance the training time and effect needs further study.

**Prompt** | *a red train, a blue bowl, a green flower, and a yellow cake*

**Janus-Pro** | **Janus-Pro w/ zero-shot semantic-level CoT** | **T2I-R1**

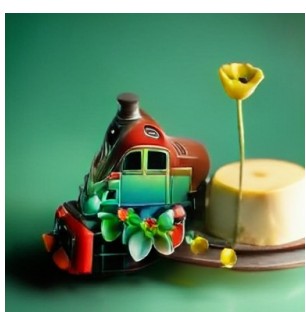 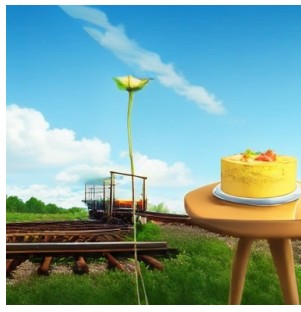 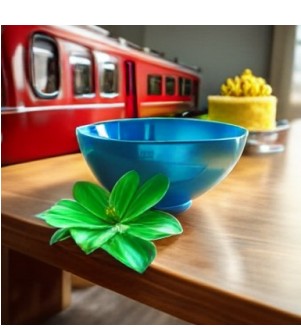

**Zero-shot Semantic-level CoT**

Underneath a blue sky, a red train is moving along the railway track. Next to it, a green flower stands tall, while a yellow cake sits on a nearby table.

**T2I-R1 Semantic-level CoT**

In the image, the red train occupies the left side of the image, with a sleek and polished metallic finish. Beside it, a blue bowl is sitting gently on a wooden table, its surface reflecting light, and there lies a delicate green flower on the table, its petals spreading like a butterfly's wings. In the background, an appetizing yellow cake with an intricate icing decoration stands next to a table on the right side of the scene. The train appears slightly ahead of the bowl, while the cake sits further behind the table and flower, creating an interesting spatial arrangement of objects and colors against a bright backdrop.

Figure 9: **Visualization Results of Semantic-level CoT.** We provide the image generation results of the same prompt from three settings: base model, base model with zero-shot semantic-level CoT, and T2I-R1. For the setting of base model with zero-shot semantic-level CoT, we use the same generation pipeline of T2I-R1 directly on the base model. We employ the same prompt of T2I-R1 to instruct the base model to generate a zero-shot semantic-level CoT, which we visualize in the figure and provide a comparison of the semantic-level CoT generated by T2I-R1.

| **Prompt** | *three paintings and one knife* |

| **Janus-Pro** | **Janus-Pro w/ zero-shot semantic-level CoT** | **T2I-R1** |

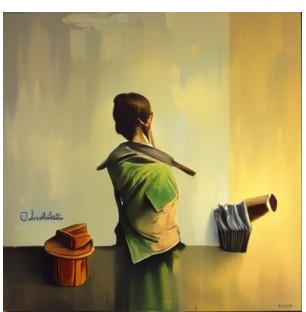 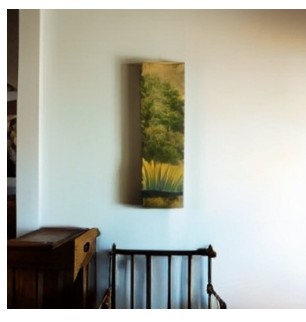 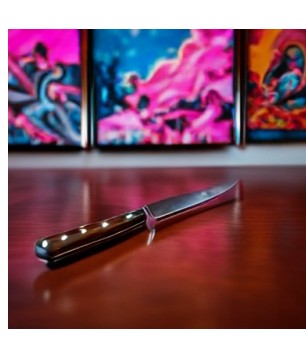

**Zero-shot Semantic-level CoT**

Three paintings hung on the wall, and in the middle stood a sharp-looking knife. The three paintings were of varying nature: one depicted sunflowers in a vibrant yellow, the second portrayed a serene countryside with soft blues and greens, while the third had an abstract, multicolored design.

**T2I-R1 Semantic-level CoT**

The image displays three paintings hung on a wall, each characterized by vibrant colors such as pink, blue, and yellow, reflecting various artistic styles. In front of the paintings, there is a single, slender knife with a metallic polish, positioned horizontally and lying on a polished wooden surface. The knife's reflective glare contrasts with the richness of the paintings, creating an intricate visual interplay. The paintings provide a backdrop to the knife, and their colors accentuate the sharpness of the blade.

Figure 10: **Visualization Results of Semantic-level CoT.** We provide the image generation results of the same prompt from three settings: base model, base model with zero-shot semantic-level CoT, and T2I-R1. For the setting of base model with zero-shot semantic-level CoT, we use the same generation pipeline of T2I-R1 directly on the base model. We employ the same prompt of T2I-R1 to instruct the base model to generate a zero-shot semantic-level CoT, which we visualize in the figure and provide a comparison of the semantic-level CoT generated by T2I-R1.

| Prompt | *a dog behind the door* |
|---|---|

| **Janus-Pro** | **Janus-Pro**
**w/ zero-shot**
**semantic-level CoT** | **T2I-R1** |
|---|---|---|

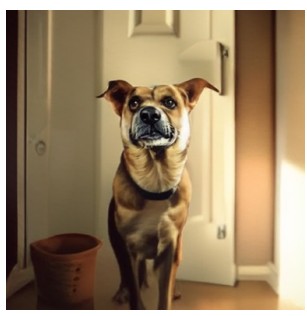 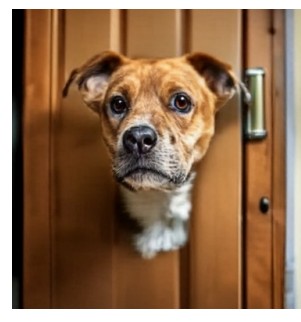 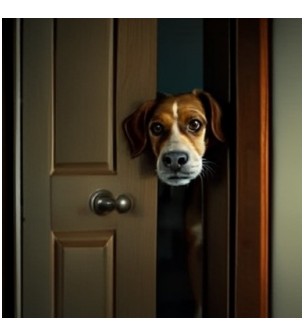

**Zero-shot**
**Semantic-level CoT**

A dog is standing behind a door, with brown fur, dark eyes, a black collar, and a white patch on its face. The door is wooden, with a metal handle and hinges. The dog's ears are perked up, and it looks curious or alert through the door.

**T2I-R1**
**Semantic-level CoT**

The image shows a brown and white dog **peeking its head through a small gap in the door**, with a gentle and curious expression etched onto its face. The door is slightly ajar, and **the dog's body is barely visible beyond the edge of the frame**, hinting at a playful waiting game. The room behind the door is dimly lit, creating an eerie yet intimate atmosphere as if the dog is on a mission or has been guarding the entrance for some time. The door itself is wooden with a simple metal handle. The overall tone of the visual emphasizes curiosity and anticipation.

Figure 11: **Visualization Results of Semantic-level CoT.** We provide the image generation results of the same prompt from three settings: base model, base model with zero-shot semantic-level CoT, and T2I-R1. For the setting of base model with zero-shot semantic-level CoT, we use the same generation pipeline of T2I-R1 directly on the base model. We employ the same prompt of T2I-R1 to instruct the base model to generate a zero-shot semantic-level CoT, which we visualize in the figure and provide a comparison of the semantic-level CoT generated by T2I-R1.

**Prompt** | *a dog, a bird, a lake, and a building*

**Janus-Pro** | **Janus-Pro w/ zero-shot semantic-level CoT** | **T2I-R1**

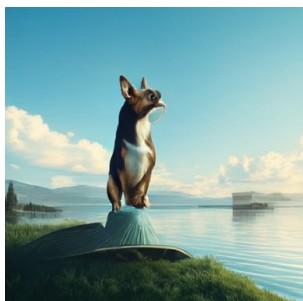 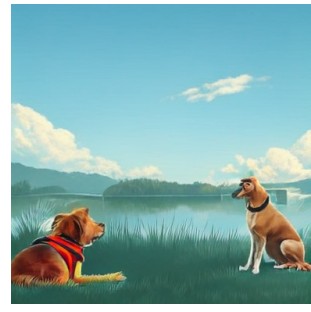 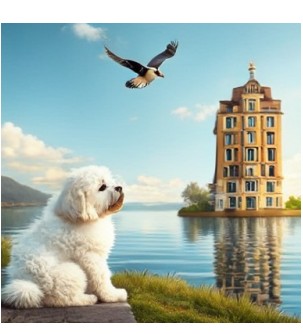

**Zero-shot Semantic-level CoT** | A dog wags its tail, plays fetch at the lake, and sits in front of the building.

**T2I-R1 Semantic-level CoT** | A fluffy white dog with bright amber eyes sits by the edge of a serene lake, observing a vibrant black and white bird soaring above. Nearby, a classic building in shades of brown and cream stands tall with numerous windows, giving it an old-world charm. Together, they create a picturesque scene that evokes peace and charm.

Figure 12: **Visualization Results of Semantic-level CoT.** We provide the image generation results of the same prompt from three settings: base model, base model with zero-shot semantic-level CoT, and T2I-R1. For the setting of base model with zero-shot semantic-level CoT, we use the same generation pipeline of T2I-R1 directly on the base model. We employ the same prompt of T2I-R1 to instruct the base model to generate a zero-shot semantic-level CoT, which we visualize in the figure and provide a comparison of the semantic-level CoT generated by T2I-R1.

**Janus-Pro**            **T2I-R1**

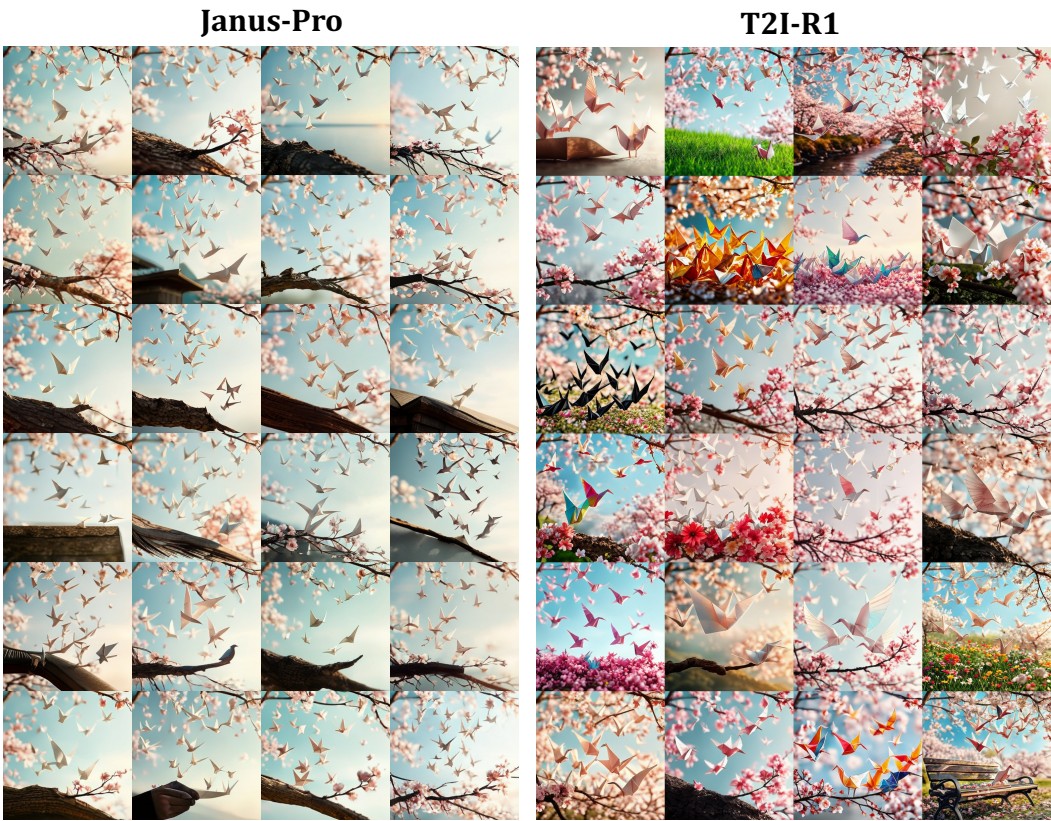

*origami cranes unfolding into real birds during cherry blossom season*

Figure 13: **More Visualization Result of the Image Diversity of a Single Prompt.** We showcase the result of the baseline model, Janus-Pro, and T2I-R1.

**Janus-Pro**              **T2I-R1**

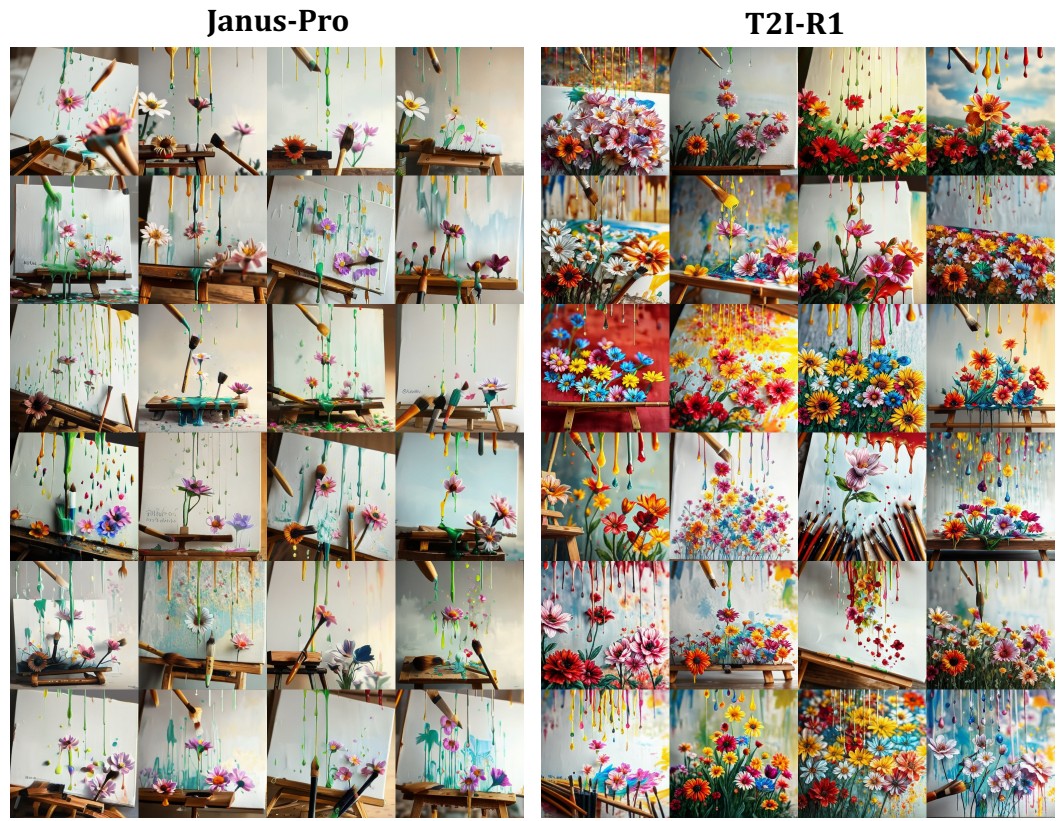

*paint drops falling from brushes creating flowers on a canvas below*

Figure 14: **More Visualization Result of the Image Diversity of a Single Prompt.** We showcase the result of the baseline model, Janus-Pro, and T2I-R1.

