# OpenReview forum: "T2I-R1: Reinforcing Image Generation with Collaborative Semantic-level and Token-level CoT"
_NeurIPS.cc/2025/Conference — NeurIPS 2025 poster_

### Official Review · Reviewer_FgvF · 2025-07-01

**Clarity:** 4
**Significance:** 3
**Originality:** 3
**Rating:** 4
**Confidence:** 4

**Summary:**

This paper introduces T2I-R1, a novel text-to-image generation model that pioneers a bi-level Chain-of-Thought (CoT) reasoning process. By effectively decoupling high-level semantic planning from low-level, patch-by-patch synthesis and optimizing both via a robust reinforcement learning framework (BiCoT-GRPO), the authors present a highly effective and well-validated method. The work's core concept is innovative, its experimental results are compelling—achieving state-of-the-art performance on multiple benchmarks.

**Questions:**

# Questions
1. About Semantic-Level CoT

For simple, unambiguous prompts (e.g., 'a red apple'), could the authors discuss how the model avoids 'over-planning'—where the generated textual details might unnecessarily constrain the model's stochastic creativity in aspects like lighting, texture, and composition?

2. About the analogy on 'Token-Level CoT'

The paper terms the patch-by-patch generation process 'Token-Level CoT,' drawing an analogy to textual CoT. Could the authors further elaborate on what distinguishes this process from standard autoregressive generation, beyond the inherent sequential dependency? For instance, textual CoT often involves intermediate steps that are human-interpretable and logically verifiable. How does this core aspect of 'thought' translate to the token-level visual generation? A deeper conceptual clarification would help solidify the paper's contribution in this regard.

3. About Visual Quality

To further strengthen the analysis on the impact of BiCoT-GRPO on Visual Quality, could the authors provide more details on the human study protocol? The paper's primary contribution is enhancing compositional reasoning. Could the authors offer a more explicit discussion on how the BiCoT-GRPO framework specifically impacts aesthetics as a distinct objective? For example, is there a risk that the RL process might overly prioritize compositional correctness at the expense of aesthetic quality in some cases, and how does the ensemble reward system mitigate this potential conflict?

**Ethical Concerns:**

["NO or VERY MINOR ethics concerns only"]

**Final Justification:**

I appreciate the authors' clarifications. I think the quality of this paper is sufficient for acceptance at NeurIPS.  I will therefore maintain my positive rating.

**Limitations:**

yes

**Quality:**

3

**Strengths And Weaknesses:**

# Strengths
1. **Novel Concept**: Introduces an innovative "bi-level Chain-of-Thought" (CoT) framework.
2. **Thorough Experiments**: Features comprehensive ablation studies validating each component.
3. **Smart Reward System**: Uses a robust ensemble of "vision experts" to guide RL training effectively.

# Weaknesses
The term "Token-Level CoT" for the standard autoregressive, patch-by-patch generation process can be seen as a loose analogy. While the authors define it based on its sequential and conditional nature the "thought" aspect is less explicit compared to the semantic-level CoT. Since there is no explicit thinking process, just a patch-by-patch image generation process.

---

> ### Author Rebuttal · Authors · 2025-07-31
>
> > **Q1 & Q3: The term "Token-Level CoT" for the standard autoregressive, patch-by-patch generation process can be seen as a loose analogy. While the authors define it based on its sequential and conditional nature the "thought" aspect is less explicit compared to the semantic-level CoT. Since there is no explicit thinking process, just a patch-by-patch image generation process. & How does this core aspect of 'thought' translate to the token-level visual generation? A deeper conceptual clarification would help solidify the paper's contribution in this regard.**
>
> Thank you for your valuable advice. We acknowledge that our terminology extends the CoT concept beyond its traditional textual reasoning context. We would like to clarify the concept of token-level CoT and address your concerns:
>
> 1. First, **we believe there must exist a type of reasoning in the process of image generation**, otherwise, we will not observe model with only token-level CoT optimized beats the baseline model on the same prompt or even in the setting where the top half image is given and the model is asked to generate the remaining ones.
>
> 2. Then, **we will revisit the definition of CoT in the textual context, and lists where the autoregressive image generation bears the similarity.**
>
>    For the textual CoT, it is defined as the intermediate natural language reasoning steps that lead to the final output in [1]. The CoT contains a long reasoning process with problem analysis and logical inference, then followed by a short final answer. It is commonly acknowledged that the part of the long output corresponds to the part of extensive thinking. The 'thought' process is reflected in two ways:
>
>
>    + Decomposition of the problem to steps
>    + Logical coherence within the step
>
>
>    Since the model outputs the tokens in the language space, this process can be easily interpreted by the human. The two aspects correspond to the logical path of solving and the correctness of each reasoning step.
>
>
>    **For the token-level CoT, the key insight is that the model's "thought process" or intermediate reasoning occurs simultaneously with final output generation, rather than as separate steps.** The model reasons and generates in parallel, integrating its thinking directly into the creation process. Different from textual CoT, the token-level CoT is much longer compared and also holds a strict sequential and conditional nature. The two 'thought' process become:
>
>
>    + The logical path of solving is determinant in the context of token-level CoT, which is the raster-scan order.
>    + The correctness of each reasoning step corresponds to the choice of specific visual tokens, which is what our method tries to optimize. This correctness includes patch coherence, object appearance, lighting conditions, and other visual details. These intermediate steps could also be verified and interpreted by humans by decoding the visual tokens into images.
>
> 3. In summary, the modality difference between autoregressive text and image generation causes the "thought" happens in the different place. We draw an analogy between the two key aspects between the textual CoT and token-level CoT. **The token-level CoT refers to the 'thought' process we identify in the autoregressive generation.** We focus on optimizing the aspect on choosing the correct visual token.
>
> We will include this clarification and comparison in our final version.
>
> [1] Wei, Jason, et al. "Chain-of-thought prompting elicits reasoning in large language models."
>
> > **Q2: About Semantic-level CoT: unambiguous prompts (e.g., 'a red apple'), could the authors discuss how the model avoids 'over-planning'—where the generated textual details might unnecessarily constrain the model's stochastic creativity in aspects like lighting, texture, and composition?**
>
> Thank you for this insightful question. We would like to address your concerns both qualitatively and quantitatively:
>
> 1. **Qualitatively (Figure 13 to 14)**
>
>    Our experiment analysis in Figure 13 and 14 in the appendix reveals that **our introduced semantic-level CoT actually enhances the diversity of the generated images**. As shown in Figure 13, the base model Janus-Pro possess relatively low diversity. The generated images of Janus-Pro are basically the same: there is a tree branch on the bottom left corner and color of the origami cranes are all white with same zoom-in scale. While T2I-R1 generates much more diverse images with different backgrounds, cranes colors, and zoom-in scales. The same conclusion holds for Figure 14.
>
> 2. **Quantitatively**
>
>    **We further conduct an experiment to validate this conclusion.** We generate 300 unambiguous prompts based on the example *'a red apple'*. We enumerate 150 common objects and 10 colors and then randomly choose 2 colors to combine with each object. Then we use the base model Janus-Pro, the model trained only with the semantic-level CoT, and T2I-R1 to generate 10 images on each prompt. Here we include the model trained only with the semantic-level CoT to eliminate the potential influence on the diversity from the token-level CoT. The results are as follows:
>
>    |Model|Diversity|
>    |-|-|
>    |Janus-Pro-7B|7.460|
>    |Only Semantic-level CoT optimized|8.612|
>    |T2I-R1|8.519|
>
>    The results clearly demonstrate that **semantic-level CoT increases output diversity by 15.4% instead of harming it**.
>
> > **Q4: To further strengthen the analysis on the impact of BiCoT-GRPO on Visual Quality, could the authors provide more details on the human study protocol? The paper's primary contribution is enhancing compositional reasoning. Could the authors offer a more explicit discussion on how the BiCoT-GRPO framework specifically impacts aesthetics as a distinct objective? For example, is there a risk that the RL process might overly prioritize compositional correctness at the expense of aesthetic quality in some cases, and how does the ensemble reward system mitigate this potential conflict?**
>
> Thanks for this important question regarding the balance between the compositional generation and aesthetic quality.
>
> 1. **Human Study Protocol**
>
>    We ask the annotators to judge only according to the visual appeal. We provide examples of an unsatisfactory generation, including different aspects:
>
>    + Corrupted floating words
>    + Distortion like melting effect of faces, hands, or other objects
>    + Extra or missing hands, legs, tails, and etc.
>    + Object merging, frequently observed for compositional prompts
>    + Low-level aspects including over exposure or over saturatation
>    + Blurred areas, especially in fine details like keyboard
>
>    The instruction to the annotator highlights that the highly ranked images should show no or less problems above compared with others.
>
> 2. **Aesthetics Quality**
>
>    We would like to address your concern of the aesthetics of the generated images:
>
>    **HPS score contains extra judgment based on the human aesthetic preferences**, as shown in Fig. 4 and Line 167. For the training data of HPS [1], the model is trained on the human preference data, which contains both the alignment and visual appealing aspects. Therefore, **the images generated by T2I-R1 actually shows better visual quality compared with the baseline model**.
>
>    We employ a different image quality judgement model to score the generated images from both Janus-Pro and T2I-R1 to further showcase the superior visual quality:
>
>    |Model|Aesthetic Score|
>    |-|-|
>    |Janus-Pro-7B|5.41|
>    |T2I-R1|5.76|
>
>    Specifically, we employ a CLIP-based aesthetic predictor, which receives the image and outputs a score. The predictor does not require prompt input so that the score is purely based on the visual appealing. We input all of the generated images when testing T2IComp-Bench to this predictor and record the average score. The result showcases that T2I-R1 improves the generation quality of the baseline model.
>
> [1] Wu, Xiaoshi, et al. "Human preference score: Better aligning text-to-image models with human preference." Proceedings of the IEEE/CVF International Conference on Computer Vision. 2023.

---

> > ### Comment · Reviewer_FgvF · 2025-08-04
> >
> > I appreciate the effort the authors have put into these clarifications and additional experiments, which I believe will significantly improve the final paper.
> >
> > Given this, I will maintain my positive rating.

---

> > > ### Author Response · Authors · 2025-08-05
> > >
> > > Thank you for your response. Your constructive insights are invaluable to our work.

---

### Official Review · Reviewer_1t3d · 2025-07-02

**Clarity:** 3
**Significance:** 3
**Originality:** 2
**Rating:** 4
**Confidence:** 4

**Summary:**

This paper introduces T2I-R1, the first reasoning-enhanced text-to-image generation model that applies chain-of-thought (CoT) reasoning to visual generation. This work bridges the gap between language model reasoning capabilities and visual generation, showing that explicit reasoning processes can significantly enhance image generation quality and prompt alignment.

**Questions:**

1. The computational overhead and complexity of the two-step generation process and ensemble reward system, which may limit practical deployment and accessibility.
2. The work lacks insights into RL training and optimization. Overall, it appears to be a straightforward application of the GRPO algorithm to text-to-image tasks, lacking fundamental innovation.

**Ethical Concerns:**

["NO or VERY MINOR ethics concerns only"]

**Final Justification:**

Thanks for the clarifications. The authors solved most of my concerns. I am willing to raise my final rating to 4 (Borderline accept).

**Quality:**

3

**Strengths And Weaknesses:**

Strengths:
1. First to identify and formalize two distinct types of reasoning (semantic-level and token-level) that mirror human artistic creation process. The dual-level approach addresses both global planning and local detail generation.
2. The proposed approach achieves significant quantitative improvements: +13% on T2I-CompBench, +19% on WISE benchmark.

Weaknesses:
1. The two-step generation process requires generating semantic COT, then image tokens, increasing inference time. Doesn't compare against using a dedicated LLM for planning + separate image generator
2. Ensemble of multiple expert models is applied to prevent reward hacking. However, it requires careful tuning of multiple reward models and their combinations. The importance of each reward model is not clear in Table 3.
3. Multi-stage training with RL makes reproduction challenging.
4. The work lacks insights into RL training and optimization. Overall, it appears to be a straightforward application of the GRPO algorithm to text-to-image tasks, lacking fundamental innovation.

---

> ### Author Rebuttal · Authors · 2025-07-31
>
> > **Q1: Two-step generation requires semantic-level CoT + image tokens, which increases inference time. Doesn't compare a dedicated llm for planning + seperate image generator.**
>
> Thanks for you valuable advice. To address your concerns, we conducted additional experiments comparing our method against dedicated planning approaches and applying prompt enrichment on other backbones:
> ||Prompt Generation Model|Image Generation Model|
> |-|-|-|
> |*Exp1*|Janus-Pro-7B|Janus-Pro-7B|
> |*Exp2*|Qwen2.5-7B-Instruct|Janus-Pro-7B|
> |*Exp3*|Qwen2.5-7B-Instruct|Show-o|
>
> We show the results on T2IComp-Bench, the aesthetic score and the inference cost below:
> ||Color|Shape|Texture|Spatial|Non-Spatial|Complex|Aesthetic|Memory (GB)|Latency (sec/sample)|
> |-|-|-|-|-|-|-|-|-|-|
> |Janus-Pro|0.6359|0.3528|0.4926|0.2061|0.3085|0.3559|5.41|15.847|6.67|
> |*Exp1*|0.6451|0.3773|0.5262|0.2298|0.3049|0.3578|5.31|16.153|7.92|
> |*Exp2*|0.7004|0.4111|0.5892|0.2522|0.3088|0.3597|5.30|30.501|8.32|
> |*Exp3*|0.6759|0.4700|0.5265|0.2603|0.3017|0.3082|-|-|-|
> |T2I-R1|0.8130|0.5852|0.7243|0.3378|0.3090|0.3993|5.76|16.657|8.57|
>
> **Key Findings:**
>
> + **Beyond Prompt Enrichment**: While dedicated planning models (*Exp1-3*) show modest improvements over the baseline, all fall significantly short of T2I-R1's performance. This demonstrates that our method's effectiveness extends well beyond simple prompt enrichment and highlights the crucial role of our joint RL optimization framework.
>
> + **Aesthetic Enhancement**: Our approach not only improves compositional capabilities across all categories but also enhances aesthetic quality (5.76 vs. baseline 5.41), indicating holistic image generation improvement.
>
> + **Excellent Trade-off**: Our method demonstrates remarkable efficiency by adding only modest computational overhead—0.8 GB additional memory and 1.9 seconds per sample—when incorporating semantic-level CoT. With an average CoT length of just 97.1 tokens, our approach achieves a 16% average improvement across benchmarks. This efficiency significantly outperforms existing RL-enhanced reasoning methods that require thousands of reasoning steps to achieve comparable performance gains (as demonstrated in Figures 2 and 3 of the DeepSeek-R1 paper [1]).
>
> We will include results from these three experiments in the final version to provide a more comprehensive comparison.
>
> [1] Guo, Daya, et al. "Deepseek-r1: Incentivizing reasoning capability in llms via reinforcement learning." arXiv preprint arXiv:2501.12948 (2025).
>
> > **Q2: Ensemble of multiple expert models requires careful tuning of multiple reward models and their combinations. The importance of each reward model is not clear in Table 3.**
>
> Thanks for pointing out! In Table 3, the weight of all the rewards are set to 1. Our preliminary experiment shows the weight of the reward model causes little influence to the final training result. Here **we provide a detailed study to illustrate the our method is stable to the weight of the reward model.** We follow the training setting of T2I-R1, where three reward models, HPS, GroundingDINO, and GIT are employed.
>
> In our experiments, we multiply the weight of each reward model by 5 respectively, while maintaing the weight of other two reward models. Apart from these three experiments, we also conduct an experiment where we compute the relative reward of each reward model inside the group, and then sum up the normalized reward as the final reward for the sample. This eliminates the mean and variance difference among the reward models (termed as *Normalized Reward*). The results are shown below:
> || Color | Shape | Texture | Spatial | Non-Spatial | Complex |
> |-|-|-|-|-|-|-|
> | *5\*HPS, 1\*others*| 0.8215 | 0.5915 | 0.7337 | 0.3051 | 0.3074 | 0.4051 |
> | *5\*GDINO, 1\*others*| 0.7951 | 0.5520 | 0.7100 | 0.3313 | 0.3104 | 0.3916 |
> | *5\*GIT, 1\*others*| 0.7972 | 0.5620 | 0.7149 | 0.3357 | 0.3111 | 0.3934 |
> | *Normalized Reward* | 0.8106 | 0.5820 | 0.7142 | 0.2940 | 0.3072 | 0.3993 |
> | T2I-R1 (*Equal Weights*)| 0.8130 | 0.5852 | 0.7243 | 0.3378 | 0.3090 | 0.3993 |
>
> The result showcases that the training procedure is relatively stable to the weights of the reward model, with an average range of 2.51% fluctuation. We acknowledege that the choice of the model weights are of great importance and we will study how to adaptively find the best weights in our future work.
>
> > **Q3: Multi-stage training with RL makes reproduction challenging.**
>
> Sorry for the confusion. We would like to clarify that the stage in the paper does not mean different training phases. **Our method requires only one stage of training.** The step shown in the Fig. 3 refers to the sequential generation process of semantic-level CoT and token-level CoT. Detailedly, since the baseline model Janus-Pro-7B does not possess the ability to generate the image start token autonomously, we employ a two-step generation method. The two steps are:
>
> 1. First step: we input the image prompt and the reasoning insturction to generate the semantic-level CoT.
>
> 2. Second step: we input the image prompt, previously generated semantic-level CoT and the image start token to obtain the token-level CoT.
>
> We will further clarify this concept and replace the word stage with step to avoid causing confusion in the final draft.
>
> > **Q4: The work lacks insights into RL training and optimization. Overall, it appears to be a straightforward application of the GRPO algorithm to text-to-image tasks, lacking fundamental innovation.**
>
> We respectfully disagree with the assessment for the following reasons:
>
> 1. **Our BiCoT-GRPO method provides the first insight into leveraging GRPO to jointly optimize the CoT from both the text and image modalities in a single iteration.**
> 2. Besides, **we propose a novel ensemble of reward models for the reward modeling in the GRPO training**. Previous methods only adopt GRPO in the understanding tasks, where the reward functions are mostly rule-based and format rewards. We first study the reward functions in GRPO for the image generation tasks. Our ensemble approach combines multiple reward perspectives to deliver holistic evaluation while preventing reward hacking.
> 3. Finally, as illustrated in Line 2-9, the main focus of our paper is to explore how what types of CoT process could be utilized in the text-to-image generation task. We identify that RL is an ideal training method for eliciting the reasoning capability of the base model to generate two types of CoT. Here, **GRPO is simply the tool that enables our investigation, much like how supervised learning enables many fundamental discoveries without being the focus itself.**
>
> > **Q5: The computational overhead and complexity of the two-step generation process and ensemble reward system, which may limit practical deployment and accessibility.**
>
> Thanks for your valuable advice. We address each aspect below:
>
> 1. **The semantic-level CoT generation introduces minimal computational overhead**, as demonstrated in the comparison below:
>
> ||Training Time Per Step (sec/step)|Inference GPU Memory (GB)|Inference Latency (sec/sample)|
> |-|-|-|-|
> | Token-level CoT only| 11.4| 15.847| 6.67 |
> | Qwen2.5-7B (planning) + Janus-Pro-7B (generation) |-| 30.501| 8.32 |
> | T2I-R1 (Semantic-level CoT + Token-level CoT)|16.3|16.657|8.57|
>
> Given the substantial improvement across benchmark results, this minor overhead is practically acceptable compared to the gains achieved.
>
> 2. **The ensemble of the reward model is only used in the training time and cost-free for the inference.**
> 3. Compared to alternative two-step generation approaches that require separate models (e.g., a dedicated LLM for planning plus an independent image generator), **our unified approach requires only a single model, effectively reducing the GPU memory requirements by half during inference**.
>
> > **Q6: The work lacks insights into RL training and optimization. Overall, it appears to be a straightforward application of the GRPO algorithm to text-to-image tasks, lacking fundamental innovation.**
>
> Please refer to our reponse to Q4.

---

> > ### Comment · Reviewer_1t3d · 2025-08-01
> >
> > The authors have solved most of my concerns. Thanks for your efforts.

---

> > > ### Author Response · Authors · 2025-08-02
> > >
> > > Thank you for your response. If you have any remaining concerns, please let us know, and we’ll respond promptly. If you feel our work merits it, we’d be grateful if you could consider raising the rating.

---

> > > > ### Comment · Reviewer_1t3d · 2025-08-02
> > > >
> > > > Yes. I already raised my rating. Considering other reviewers comments, I think this work is ok to accept to the conference.

---

> > > > > ### Author Response · Authors · 2025-08-02
> > > > >
> > > > > We are grateful for your detailed feedback and the improved score. Your constructive insights are invaluable to our work.

---

### Official Review · Reviewer_7zUq · 2025-07-02

**Clarity:** 3
**Significance:** 2
**Originality:** 3
**Rating:** 5
**Confidence:** 4

**Summary:**

This paper proposes T2I-R1, a reasoning-enhanced text-to-image generation framework that integrates reinforcement learning (RL) with bi-level Chain-of-Thought (CoT) reasoning. It introduces two levels of CoT: semantic-level CoT for high-level prompt planning, and token-level CoT for fine-grained pixel-wise generation. To unify these two reasoning processes, the authors design BiCoT-GRPO, a training mechanism with an ensemble of generation rewards to jointly optimize both levels. Applied to the Janus-Pro baseline, T2I-R1 achieves significant performance improvements, outperforming prior state-of-the-art methods by 13% on T2I-CompBench and 19% on WISE.

**Questions:**

see Weakness.

**Ethical Concerns:**

["NO or VERY MINOR ethics concerns only"]

**Final Justification:**

The authors have provided a thorough rebuttal with detailed responses and additional experiments that effectively address all of my concerns. The proposed approach is novel and well-motivated. I recommend acceptance.

**Limitations:**

see Weakness.

**Paper Formatting Concerns:**

No.

**Quality:**

3

**Strengths And Weaknesses:**

Strengths

1. The paper introduces a novel bi-level reasoning paradigm for image generation, distinguishing between semantic-level and token-level CoT processes.

2. It proposes a reward-driven optimization strategy for RL-based image generation, which is well-aligned with the model's hierarchical reasoning structure.

3. The approach is empirically validated on multiple benchmarks, demonstrating performance improvements over strong baselines.


Weaknesses

1. The semantic-level CoT aims to enrich the initial prompt by imagining more detailed aspects of the target image. However, this idea closely relates to prior work on prompt enrichment in text-to-image generation, which is not discussed.

- Mastering Text-to-Image Diffusion: Recaptioning, Planning, and Generating with Multimodal LLMs
- Prompt Refinement with Image Pivot for Text-to-Image Generation
- DisEnvisioner: Disentangled and Enriched Visual Prompt for Customized Image Generation
- Paragraph-to-Image Generation with Information-Enriched Diffusion Model

2. The paper assumes a patch-by-patch generation process similar to auto-regressive models and maps this to token-level CoT. However, this assumption may not hold for diffusion-based generation, which follows a denoising trajectory rather than token-wise generation. Whether token-level CoT is meaningful or effective under diffusion paradigms is unclear. Alternative strategies, such as coarse-to-fine generation (e.g., as in VAR or Next-Scale Prediction), might be more appropriate and should be discussed.

- Scalable Image Generation via Next-Scale Prediction

3. The role of reinforcement learning in this framework needs to be further justified. It’s unclear whether RL is essential for the proposed model or whether similar gains could be achieved through supervised or hybrid optimization.

4. Experimental results suggest that most performance gain comes from the semantic-level CoT. However, it remains unclear how much of this improvement comes purely from prompt enrichment, without involving RL or token-level reasoning. A fairer comparison would include a model with enriched prompts but no further optimization, especially in the context of diffusion-based generators.

---

> ### Author Rebuttal · Authors · 2025-07-31
>
> > **Q1: Semantic-level CoT is to enrich the initial prompt by adding more details. However, this idea is closely related to prompt enrichment, which is not discussed.**
>
> Thanks for pointing out. After thoroughly reading the related papers mentioned, **we acknowledge that the idea of semantic-level CoT is related to the prompt enrichment work**. However, we would like to **highlight some key differences and advantages between our method and the prompt enrichment technique**:
>
> 1. The semantic-level CoT is generated from the image generation model itself, while the prompt enrichment leverages an extra LLM. **Our design facilitates a joint-optimization of the prompt design and the image generation. On the contrary, the prompt enrichment is generation-model-agnostic**, which means that although the prompt is enriched with more details, it is not necessary that the generation model could generate a better image based on this enriched prompt.
>
>    To validate this claim, we conduct the following experiments on T2IComp-Bench:
>
>    ||Prompt Generation Model|Image Generation Model|
>    |-|-|-|
>    |*Exp1*|Janus-Pro-7B|Janus-Pro-7B|
>    |*Exp2*|Qwen2.5-7B-Instruct|Janus-Pro-7B|
>
>    Note that, T2I-R1 is obtained after *Exp1* is trained with our proposed BiCoT-GRPO. The results are shown below:
>
>    ||Color|Shape|Texture|Spatial|Non-Spatial|Complex|Aesthetic|
>    |-|-|-|-|-|-|-|-|
>    |Janus-Pro|0.6359|0.3528|0.4926|0.2061|0.3085|0.3559|5.41|
>    |*Exp1*|0.6451|0.3773|0.5262|0.2298|0.3049|0.3578|5.31|
>    |*Exp2*|0.7004|0.4111|0.5892|0.2522|0.3088|0.3597|5.30|
>    |T2I-R1|0.8130|0.5852|0.7243|0.3378|0.3090|0.3993|5.76|
>
>    As shown, using the prompt enriched from either Qwen2.5 or Janus-Pro-7B itself still far lag behind T2I-R1 in both compositional abilities and aesthetics.
>
>    Specifically, to evaluate aesthetics, we employ a CLIP-based aesthetic predictor, which receives the image and outputs a score. The predictor does not require prompt input so that the score is purely based on the visual appealing. We input all of the generated images when testing T2IComp-Bench to this predictor and record the average score.
>
> 2. **Qualitative examples shown in Figure 9 to 12 also shows that our method goes beyond prompt enrichment.** Although the zero-shot semantic-level CoT (enriched prompt) in Fig. 9 to 11 all correctly include all the key objects mentioned in the prompt and add more details, the model still cannot generate satisfying result. Simply enriching the prompt is not sufficient to bring notable improvements.
>
> 3. Since the semantic-level CoT and the image are generated from the same model, **the deployment cost is much smaller than the prompt enrichment method:**
>
>    || Memory (GB)|
>    |-|-|
>    |T2I-R1|16.657|
>    |*Exp2*|30.501|
>
> We will include the discussion of prompt enrichment and cite all the works you mentioned in the related work in our final version.
>
> > **Q2: Whether token-level CoT is meaningful under diffusion paradigms is unclear. Alternative strategies, such as VAR should be discussed.**
>
> Thank you for raising this important point. **As illustrated in Line 27, the focus of our paper is the autoregressive text-to-image generation**, where the images are generated token by token in a discrete space.
>
> + **For the diffusion-based models**, they employ iterative denoising in continuous space that are totally different from our settings. While some notable parallels exist (e.g., the generation process of diffusion model resembles a type of CoT happening in the noise space instead of the token space), this would require a complete theoretical reformulation. We will leave this for future work and will further clarify our focus in the introduction.
>
> + **As for VAR**, we believe our method could be directly applied. Techinically, VAR also adopts an autoregressive generation manner and the generation image encoder is also discrete.  The only difference is that VAR first generates the coarse tokens then the fine-grained ones, but this does not conflict with our motiavation or method. In fact, it is interesting that this hierarchical reasoning from the coarse structure to fine details perfectly matches the spirit of CoT reasoning. We will also include this discussion into the future work part in our final version.
>
> > **Q3: Role of RL should be justified. It's unclear whether RL is essential or whether similar gains could be achieved through supervised or hybrid optimization.**
>
> Thanks for your advice. **We conduct a supervised finetuning (SFT) and a hybrid optimization (SFT then RL) to validate the advantage of direct RL training.**
>
> + **SFT:** We first source 200K high-quality image-text pairs. Then we use a MLLM to generate a detailed thinking process based on the image and original short prompt, which is used as the semantic-level CoT in the SFT training. During training, we jointly train two tasks together:
>
>   1. Input: Image Prompt + Reasoning Instruction; Output: Semantic-level CoT
>   2. Input: Image Prompt + Semantic-level CoT + <img_start>; Output: Token-level CoT.
>
>   This design strictly follows the generation manner of T2I-R1 for a fair comparison. We train the model for one epoch with learning rate of 1e-5.
>
> + **Hybrid Optimization:** After the SFT training, we follow the same setting of T2I-R1 to train the SFT model with reinforcement learning.
>
> The results are shown below:
>
> ||Color|Shape|Texture|Spatial|Non-Spatial|Complex|
> |-|-|-|-|-|-|-|
> |SFT|0.6385|0.4698|0.5429|0.1950|0.2987|0.3305|
> |Hybrid optimization|0.8003|0.5764|0.7012|0.3011|0.3082|0.3878|
> |T2I-R1 (RL)|0.8130|0.5852|0.7243|0.3378|0.3090|0.3993|
>
> **Key Findings:**
>
> + **Both SFT and hybrid optimization fails to outperform training only with RL.** This likely stems from the model's existing proficiency of generating semantic-level and token-level CoT. The additional SFT process forces the model to learn from data that is misaligned with its current capabilities, resulting in minimal or even detrimental improvements.  Furthermore, SFT constrains the model's generation capabilities to match the training data distribution, reducing output diversity and limiting the exploration space during the subsequent RL phase.
>
> + **The training cost of SFT and hybrid optimization are also larger than exclusive RL training.** We showcase the total training hours and the data that needs annotation. SFT requires annotating semantic-level CoT based on existing text-image pairs datasets while RL needs no data annotation.
>
>   ||Total Training Hours|Needed Annotation|
>   |-|-|-|
>   |SFT|4.3|Semantic-level CoT|
>   |Hybrid optimization|29.0|Semantic-level CoT|
>   |T2I-R1 (RL)|14.5|None|
>
> We will include this experiment and the corresponding conclusion in our final version to stength the role of reinforcement learning.
>
> > **Q4: Most performance comes from semantic-level CoT. It remains unclear how much of this improvement comes purely from prompt enrichment, without involving RL or token-level reasoning. A fairer comparison would include a model with enriched prompts but no further optimization, especially in the context of diffusion-based generators.**
>
> Thanks for your valuable advice. We want to address your concerns from three aspects:
>
> 1. **Token-level CoT contributes substantially to performance improvement.** We show the benefit obtained from token-level CoT from two aspects:
>
>    1. **Qualitative Aspect (Figure 5)**
>
>       The ablation visualizations in Figure 5 clearly demonstrate token-level CoT's crucial role. . Comparing columns 2 (semantic-level CoT only) and columns 4 (semantic-level and token-level CoT), the improvement brought by the token-level CoT is obvious. For example, with the top left prompt *"Show a plant that is asymbol of good fortune inIrish culture, and is knownfor its three-lobed leaves"*:
>
>       + **Semantic-level only (Column 2)**: Correctly identifies "clover" but produces a flat, textureless image with poor lighting and plain white background
>       + **With token-level CoT  (Column 4)**: Generates vibrant colors, natural lighting, realistic textures, and professional depth-of-field that makes the subject prominent
>
>    2. **Quantitative Aspect (Table 4)**
>
>       We also illustrate the effectiveness of the token-level CoT quantitatively.
>
>       First, **for the compositional ability** in Tabel 4, it can be shown that only optimising the token-level CoT brings large improvement, with 20% on T2I-CompBench and 5% on WISE benchmark.
>
>       Second, **we provide a quantitative metric to measure the improvement of visual quality brought by the token-level CoT:**
>
>       |Model|Semantic-level CoT|Token-level CoT|Aesthetic Score|
>       |-|-|-|-|
>       | Janus-Pro-7B |||5.41|
>       |-|$\checkmark$|| 5.58|
>       |-||$\checkmark$|5.68|
>       |T2I-R1|$\checkmark$|$\checkmark$|5.76|
>
> 2. **Ablation of Prompt Enrichment.**
>
>    The qualitative results from Figure 9 to 12 in the appendix showcase that **a valid prompt enrichment (zero-shot semantic-level CoT in the figure) does not necessarily produce aligned images**. Without learned coordination between planning and generation, the model fails to follow the enriched prompt.  In contrary, by our optimization, T2I-R1 could leverage and understand this enriched prompt to generate satisying results.
>
>    The quantitative results from *Exp1* and *Exp2* have also shown that the improvement of our method has only partly come from the prompt enrichment.
>
> 3. While we focus on autoregressive models, we understand the reviewer's interest in broader applicability. However, **testing prompt enrichment on diffusion models seems to provide very limited insights**. There are huge differences including theoretical foundations, architecture, and training method between these two kinds of models. This cross-paradigm results wouldn't validate or invalidate our autoregressive findings.
>
>    Nevertheless, we're open to discussing specific experimental designs if the reviewer believes certain diffusion experiments would strengthen our analysis.

---

> > ### Comment · Reviewer_7zUq · 2025-08-02
> >
> > Thank the authors for the detailed response. After a careful review, I still have a few minor questions:
> >
> > 1. When using Janus-Pro as the prompt generation model, its performance appears to be lower than that of Qwen2.5-7B-Instruct. Could the authors provide further experimental insights into this discrepancy? For instance, are there notable differences in the quality or characteristics of the prompts generated by the two models?
> >
> > 2. If the model already exhibits proficiency in generating semantic-level chain-of-thought (CoT) reasoning, why does additional supervised fine-tuning (SFT) lead to misalignment with its existing capabilities? Is the observed diversity in model outputs due to inherent weaknesses in semantic-level CoT reasoning, resulting in more randomness? Additionally, the authors mention using 200K SFT examples—would employing a smaller but higher-quality SFT dataset to enhance semantic-level CoT generation, combined with reinforcement learning, potentially yield better alignment and performance?
> >
> > I would appreciate the authors' further discussion on these points.

---

> > > ### Author Response · Authors · 2025-08-04
> > >
> > > > **Q1: Further experimental insights into the discrepancy between the semantic-level CoT generated by Janus-Pro and Qwen2.5-7B-Instruct.**
> > >
> > > Thank you for your advice. We conduct a comprehensive evaluation to understand the performance differences between using Janus-Pro and Qwen2.5-7B-Instruct as the semantic-level CoT generation model:
> > >
> > > ### **Quantitatively**
> > >
> > > We first define what makes a good semantic-level CoT based on the following four criteria:
> > >
> > > 1. **Alignment (20 points total)**
> > >
> > > Evaluate how faithfully the semantic-level CoT represents the original prompt:
> > >
> > > - **Existence (5 points)**: All objects mentioned in the original prompt appear in the semantic-level CoT
> > > - **Attributes (5 points)**: All specified object attributes (colors, materials, sizes, etc.) are preserved in the semantic-level CoT. If no attributes are specified in the original, award full points
> > > - **Relationships (5 points)**: All spatial relationships and interactions between objects are maintained. If no relationships are specified in the original, award full points
> > > - **Focus (5 points)**: The semantic-level CoT centers on the original prompt's content with minimal irrelevant additions
> > >
> > > 2. **Details (5 points)**
> > >
> > > Evaluate whether the semantic-level CoT provides sufficient descriptive detail for effective image generation (composition, interaction, environment, etc.)
> > >
> > > 3. **Purity (5 points)**
> > >
> > > Evaluate whether the semantic-level CoT contains only image descriptions. Failure examples:
> > >
> > > - Header text ("Here is a visualization:", "I have generated:")
> > > - Instructions or explanations
> > > - Non-descriptive content
> > > - Code, links, or formatting characters
> > >
> > > 4. **Clarity (5 points)**
> > >
> > > Evaluate whether the semantic-level CoT is well-written, coherent, and comprehensible rather than fragmented or confusing
> > >
> > > Then, we instruct Qwen2.5-32B-Instruct to score the model-generated semantic-level CoT based on this criteria. The average scores of Janus-Pro and Qwen2.5-7B-Instruct are as follows:
> > >
> > > || Existence | Attributes | Relationships | Focus | Details | Purity | Clarity | Mean |
> > > | - | - | - | - | - | - | - | - | - |
> > > | Janus-Pro           | 4.84      | 3.59       | 4.04          | 4.04  | 4.00    | 3.84   | 4.43    | 4.11 |
> > > | Qwen2.5-7B-Instruct | 4.99      | 3.84       | 4.77          | 4.95  | 4.57    | 4.99   | 4.97    | 4.73 |
> > >
> > > As shown, Qwen2.5-7B-Instruct exhibits superior stability across all metrics. Specially, we find Qwen2.5-7B-Instruct excels in the aspect of purity, highlighting its extraordinary ability of instructions following, where we require the model directly output the answer without any other output.
> > >
> > > ### **Qualitatively**
> > >
> > > We also provide some typical failure examples of Janus-Pro:
> > >
> > > ```
> > > Example 1:
> > > Original Prompt: a blue backpack and a red book
> > > Generated Semantic-level CoT: I have generated the following visualization as per your instructions:\n\n1. There is 1 blue backpack and 1 red book\n2. The blue backpack is a rectangular prism (spiked dots on its top side): the color blue\n3. The red book is a square shape (corner dots): the color red\n4. The items are placed side by side: a small distance between them
> > >
> > > Example 2:
> > > Original Prompt: "a painting on the top of a rabbit
> > > Generated Semantic-level CoT: Below is a visualization of the prompt \"a painting on the top of a rabbit\":\n\n![A painting on a rabbit (top)](xxx/rabbit-painting-top-image).\n\nThe visual depicts a beautiful, small painting placed on the top of a rabbit. The painting features bright colors and intricate patterns, adding a touch of elegance and grace to the rabbit's head. The rabbit, with its soft fur and detailed whiskers, is pictured as an adorable, lifelike creature. The painting and rabbit create a visually appealing and harmonious piece, portraying a delightful and creative scenario.
> > > ```
> > >
> > > Examples of Qwen2.5-7B-Instruct:
> > >
> > > ```
> > > Orignal Prompt: a diamond cufflink and a spherical shirt button
> > > Generated Semantic-level CoT: A diamond cufflink sparkles next to a single spherical shirt button. The diamond cufflink is rectangular with a vivid sparkly cut, while the button is glossy and round, pure white with a smooth surface.
> > >
> > > Orignal Prompt: A person is looking at a rainbow and marveling at its beauty.
> > > Generated Semantic-level CoT: A person standing outdoors is gazing upwards at a vivid, multi-colored rainbow arching in the sky. The rainbow consists of seven distinct colors: red, orange, yellow, green, blue, indigo, and violet. The person's face, showing amazement, is partially illuminated by the sunlight filtering through the rainbow's colors.
> > > ```
> > >
> > > As shown, Janus-Pro tends to output header text or even links in the semantic-level CoT. While the semantic-level CoT from Qwen2.5-7B-Instruct showcases better purity.
> > >
> > > In summary, although Janus-Pro could generate valid semantic-level CoT, it suffers from instability issues like frequent header text generation. These output confuses the model and causes unsatisfying generation.

---

> > > > ### Author Response · Authors · 2025-08-04
> > > >
> > > > >  **Q2: Why does additional SFT lead to misalignment? Additionally, would employing a smaller but higher-quality SFT dataset to enhance semantic-level CoT generation, combined with RL, potentially yield better performance?**
> > > >
> > > > Thank you for your advice. We conduct the following experiments:
> > > >
> > > > We use the open-source high-quality text-to-image dataset BLIP3o-60k as our SFT dataset. We employ Qwen2.5-VL-72B-Instruct to generate the semantic-level CoT with the image and the original short prompt input. We employ the same criteria in the Question 1 to ensure the high quality of the generated semantic-level CoT. We finetune the model for one epoch and then conduct RL finetuning following the same training setting of T2I-R1.
> > > >
> > > > |                     | Color  | Shape  | Texture | Spatial | Non-Spatial | Complex |
> > > > | ------------------- | ------ | ------ | ------- | ------- | ----------- | ------- |
> > > > | Janus-Pro           | 0.6359 | 0.3528 | 0.4926  | 0.2061  | 0.3085      | 0.3559  |
> > > > | SFT                 | 0.7035 | 0.5217 | 0.6423  | 0.2775  | 0.3068      | 0.3626  |
> > > > | Hybrid optimization | 0.7765 | 0.5832 | 0.6981  | 0.3327  | 0.3092      | 0.3949  |
> > > > | T2I-R1 (RL)         | 0.8130 | 0.5852 | 0.7243  | 0.3378  | 0.3090      | 0.3993  |
> > > >
> > > > **Key Findings:**
> > > >
> > > > 1. **High-quality SFT is beneficial to the training.** The curated BLIP3-60k dataset yields significant performance gains compared to our previous 200k dataset, demonstrating that SFT quality is crucial for consistent semantic-level CoT generation. However, SFT performance still substantially lags behind RL training. We hypothesize this occurs because SFT constrains the model to replicate the training distribution rather than leveraging its inherent capabilities. Specifically, when presented with a valid semantic-level CoT, SFT forces the model to generate the exact corresponding training image, even when the model could produce alternative valid outputs. This constraint introduces unnecessary training complexity.
> > > >
> > > > 2. **RL after high-quality SFT is still inferior to direct RL training.** While high-quality SFT pre-training improves subsequent RL performance, the combined approach remains mostly inferior to or merely comparable with direct RL training.
> > > >
> > > >    To understand this phenomenon, we provide a further analysis of estimating the potential of GRPO. Specifically, we use the same two-step generation techinque on Janus-Pro and the model after SFT to generate 8 different images based on a single prompt. Then we use a reward model, HPS, to score the images. We compute the standard deviation of the scores of the same prompt. The scale of this standard deviation reflects the exploration range. Higher standard deviation indicates greater exploration capacity, suggesting better potential for discovering high-quality samples and achieving higher performance upper bounds.
> > > >
> > > >    |           | Std  |
> > > >    | --------- | ---- |
> > > >    | Janus-Pro | 3.1% |
> > > >    | SFT       | 2.1% |
> > > >
> > > >    This suggests that the distributional constraints imposed by SFT may limit the optimization of the later RL training. Directly applying RL is the optimal choice both from the perspective of performance and training cost.

---

> > > > > ### Comment · Reviewer_7zUq · 2025-08-05
> > > > >
> > > > > Thanks for the authors' additional detailed response, and all my concerns are well addressed. I will raise my score to a positive one.

---

> > > > > > ### Author Response · Authors · 2025-08-05
> > > > > >
> > > > > > We are grateful for your detailed feedback and the improved score. Your constructive insights are invaluable to our work.

---

### Official Review · Reviewer_ypQJ · 2025-07-02

**Clarity:** 3
**Significance:** 3
**Originality:** 3
**Rating:** 5
**Confidence:** 3

**Summary:**

This paper introduces T2I-R1, which aims at performing text-to-image generation with bi-level CoT reasoning process. Specifically, the semantic-level CoT captures high-level semantics while the token-level CoT is for low-level information modelling. The proposed method BiCoT-GRPO boosts the performance of Janus-Pro by 13% on T2I-CompBench. The ablation verifies the effectiveness of different levels of the CoT reasoning process.

**Questions:**

The topic of the paper is worth investigating, and the paper well illustrates the benefit of RL in text-to-image generation. While some weaknesses above, I still think this paper is acceptable.

**Ethical Concerns:**

["NO or VERY MINOR ethics concerns only"]

**Final Justification:**

I appreciate the detailed rebuttal, and my concerns have been addressed. Thanks for the effort, and I will maintain my original score.

**Limitations:**

yes.

**Paper Formatting Concerns:**

no.

**Quality:**

3

**Strengths And Weaknesses:**

**Strengths**
1) The motivation is clear, and the exploration of RL into text-to-image generation is worthwhile. The proposed methods sound technically correct. The CoT reasoning-based generation helps improve fidelity significantly.
2) The BiCoT-GRPO unifies the semantic-level CoT and low-level CoT in a sequential manner, which leverages the semantic-enriched prompt for generation, exhibiting high quality.
3) The ensemble of generation reward models, including the object detector, VQA, HPS, and ORM helps comprehensively improve the generation quality in different aspects.

**Weakness**
1) In Table 3, I think the evaluation of the object detector on T2I-CompBench can not fully showcase the benefit of the reward design. I suppose the metrics counting in GenEval can better illustrate its effectiveness.
2) In Figure 5, if only included token-level CoT is included, the generated image is relatively similar to the original Janus-Pro output. It seems like the semantic-level CoT benefits more.
3) The function of the human Preference Model and Output Reward Model seems to overlap.

---

> ### Author Rebuttal · Authors · 2025-07-31
>
> > **Q1: The evaluation of obj detector cannot be fully showcased in T2I-CompBench. Need metrics of counting in Geneval to better illustrate.**
>
> Thanks for your advice. **We provide the results comparison of T2I-R1 with the baseline model on the counting tasks in three different benchmarks**: GenEval, T2IComp-Bench++, and GenAI-Bench:
>
> ||Baseline|T2I-R1|
> |-|-|-|
> |T2IComp-Bench++-numeracy|0.4132|0.6097|
> |GenEval-counting|0.5375|0.5688|
> |GenAI-counting|0.73|0.81|
>
> As the results shown, **T2I-R1 largely surpasses the baseline model on all of the three benchmarks.** We believe this could validate the effectiveness of the object detector in counting tasks. Note that, we cannot reproduce the result of the baseline model Janus-Pro-7B in their paper. For fair comparison, we evaluated its performance on our own, ensuring that both the baseline model and T2I-R1 used the same random seed during generation.
>
> > **Q2: In Fig. 5, token-level CoT is similar to the baseline output. It seems like the semantic-level CoT benefits more.**
>
> Sorry for the confusion. We would like to address your concerns from the following two aspects:
>
> 1. **Qualitative Aspect (Figure 5)**
>
>    The prompt in Fig. 5 are sourced from WISE, which requires a strong reasoning capability. We do not inlcude this kind of data in our training dataset. Therefore, without the semantic-level CoT helping analyze the prompt, this prompt is out of distribution for the model, causing the baseline model and model with only token-level CoT optimised generating similar non-sense images.
>
>    To observe the gain brought by the token-level CoT, **we should include the semantic-level CoT when comparing, i.e., comparing the second and forth column in Fig. 5**. The improvement brought by the token-level CoT is obvious. For example, with the top left prompt *"Show a plant that is asymbol of good fortune inIrish culture, and is knownfor its three-lobed leaves"*:
>
>    + **Semantic-level only (Column 2)**: Correctly identifies "clover" but produces a flat, textureless image with poor lighting and plain white background
>    + **With token-level CoT (Column 4)**: Generates vibrant colors, natural lighting, realistic textures, and professional depth-of-field that makes the subject prominent
>
> 2. **Quantitative Aspect (Table 4)**
>
>    We also illustrate the effectiveness of the token-level CoT quantitatively.
>
>    First, **for the compositional ability** illustrated in Tabel 4, it can be shown that only optimising the token-level CoT brings large improvement, with 20% on T2I-CompBench and 5% on WISE benchmark.
>
>    Second, **we provide a quantitative metric to measure the improvement of visual quality brought by the token-level CoT.** Specifically, we employ a CLIP-based aesthetic predictor, which receives the image and outputs a score. The predictor does not require prompt input so that the score is purely based on the visual appealing. We input all of the generated images when testing T2IComp-Bench to this predictor and record the average score.
>
>    |Model|Semantic-level CoT|Token-level CoT|Aesthetic Score|
>    |-|-|-|-|
>    | Janus-Pro-7B |||5.41|
>    |-|$\checkmark$||5.58|
>    |-||$\checkmark$|5.68|
>    |T2I-R1|$\checkmark$|$\checkmark$|5.76|
>
>    The improvement from Line 2 to Line 4 and from Line 1 to Line 3 both demonstrate the effectiveness of token-level CoT in visual quality. We will include the illustration of Fig. 5 and the aesthetic score in the final version to make the effectiveness of token-level CoT more obvious.
>
> > **Q3: Function of HPS and ORM is overlap**
>
> Sorry for the confusion. While both models evaluate prompt alignment, they also serve other distinct purposes:
>
> 1. ORM focuses exclusively on prompt-image alignment, whereas **HPS incorporates human aesthetic preferences alongside alignment assessment**, as demonstrated in Figure 4 and Line 167. For the training data of ORM, the label of an image is *Yes* whenever it is aligned with the prompt. For HPS, the model is trained on the human preference data [1], which contains both the alignment and visual appealing aspects.
> 2. **ORM leverages multimodal large language models, thereby preserving extensive world knowledge**, while HPS relies on CLIP-based architectures. With the knowledge from LLM, ORM is expected to provide a more faithful score when encourtering a unusual prompt or generated image.
>
> We will clarify these distinctions in the text in our final version.
>
> [1] Wu, Xiaoshi, et al. "Human preference score: Better aligning text-to-image models with human preference." Proceedings of the IEEE/CVF International Conference on Computer Vision. 2023.

---

> > ### Comment · Reviewer_ypQJ · 2025-08-03
> >
> > I appreciate the detailed rebuttal, and my concerns have been addressed. Thanks for the effort, and I will maintain my original score.

---

> > > ### Author Response · Authors · 2025-08-03
> > >
> > > Thank you for your response. Your constructive insights are invaluable to our work. If you have any remaining concerns, please let us know, and we’ll respond promptly.

---

### Note · Authors · 2025-08-13

We sincerely thank all reviewers for their thoughtful feedback and constructive engagement throughout our comprehensive discussion period. Their insights have been invaluable in strengthening our work.

## Review Highlights

We are pleased to learn that the reviewers have recognized and appreciated the contributions made in our work, which include:

1. **Novel ideas of dual levels of CoT for image geneartion** (Reviewer 7zUq, 1t3d, FgvF)
2. **Innovative RL-based framework to optimize two levels of CoT** (Reviewer ypqj, 7zUq)
3. **Comprehensive reward system design** (Reviewer ypqj, 7zUq, FgvF)
4. **Notable performance improvements** (Reviewer ypqj, 7zUq, 1t3d)
5. **Thorough ablation studies** (Reviewer FgvF)

## Summary of Paper Updates

Based on reviewer feedback, we are committed to incorporating the following content to strengthen our paper for the final version:

**Method**:

+ Enhanced conceptual clarification of Token-level CoT
+ Distinct role of HPS and ORM

**Experiment**:

+ Performance improvements on counting tasks
+ Clarification of qualitative and quantitaive improvements from Token-level CoT
+ Advantage over methods with prompt rewriting
+ Advantage of proposed RL training over SFT
+ Clarification of improved diversity brought by semantic-level CoT
+ Stability of the reward importance weight
+ More details of human study

**Future Work**:

+ Discussion of applications on different paradigms like diffusion models or VAR

As claimed in our paper, we will open-source all training datasets, code, and model weights to facilitate future research. We sincerely thank the Area Chair and Senior Area Chair for their time and efforts in organizing the review process.

---

### Decision · Program_Chairs · 2025-09-17

**Decision:**

Accept (poster)

**Comment:**

This paper introduces T2I-R1, a reasoning-enhanced text-to-image generation model integrating reinforcement learning with bi-level Chain-of-Thought (CoT) reasoning: semantic-level for high-level planning and token-level for low-level generation, optimized via BiCoT-GRPO. Reviewers praised its novel bi-level framework, effective reward ensemble, and significant performance gains (13% on T2I-CompBench, 19% on WISE). However, concerns were raised: under-discussed connections to prior prompt enrichment work, ambiguity in token-level CoT’s relevance to diffusion models, RL’s necessity, computational overhead, and limited insights into reward model tuning. The authors addressed key concerns in their rebuttal with clarifications and additional experiments. Despite minor limitations, T2I-R1 makes valuable contributions by bridging language reasoning and visual generation, with strong empirical support. The AC concurs this is a worthwhile advancement and recommends acceptance.